# FDN: Interpretable Spatiotemporal Forecasting with Future Decomposition Networks

## Abstract

Spatiotemporal systems comprise a collection of spatially distributed yet interdependent entities each generating unique dynamic signals. Highly sophisticated methods have been proposed in recent years delivering state-of-the-art (SOTA) forecasts but few have focused on interpretability. To address this, we propose the Future Decomposition Network (FDN), a novel forecast model capable of (a) providing interpretable predictions through classification (b) revealing latent activity patterns in the target time-series and (c) delivering forecasts competitive with SOTA methods at a fraction of their memory and runtime cost. We conduct comprehensive analyses on FDN for multiple datasets from hydrologic, traffic, and energy systems demonstrating its improved accuracy and interpretability.

## 1 Introduction

A spatiotemporal system represents a collection of spatially distributed but interdependent entities each with unique activity (Li et al., 2017; Zeng et al., 2023). This activity, such as traffic flow, is driven by a complex set of interactions resulting in emergent behaviors that are difficult to understand from observed data. In the case of traffic systems, traffic congestion generally coincides with high traffic volume and network bottlenecks. We observe similar dynamics in streamflow networks where high streamflow events and subsequent dissipation regularly coincide with major precipitation.

In this paper, we propose that system behavior can largely be explained by a finite set of fundamental activity patterns. In the context of spatiotemporal learning, a pattern represents a temporal signature of the target variable that recurs frequently and follows specific system rules. For instance, the rise and fall of streamflow during flood events follows major precipitation, hence, we can expect to observe this pattern during similar weather events. These temporal patterns resemble filters used in image processing to detect specific features (Krizhevsky et al., 2012).

We propose a model that aims to detect these recurring patterns, as they are likely to reappear in the future. However, future patterns may not precisely match past ones. Therefore, we frame the problem as a soft classification task, estimating the probability of different patterns contributing to the forecast. Using the classification probabilities, we interpolate from a set of learned patterns to make final predictions. We refer to this approach as the Future Decomposition Network (FDN): a model which decomposes system activity (the training data) into important patterns, (softly) classifies past activity, and predicts the future as an interpolation of these patterns.

As evidence, we can represent a system of $N$ entities containing $B$, $O$-time-step patterns as a matrix $\mathbb{F} \in \mathbb{R}^{N \cdot O \times B}$. While $\mathbb{F}$ captures all known system behavior, it is highly redundant and may be closely reproduced by a small set of $K$ fundamental patterns $\hat{\mathbb{F}} \in \mathbb{R}^{O \times K}$ shared by all entities. For example, in the Wabash River data analyzed in this paper, 70 years of localized streamflow across 1,276 subbasins can be effectively represented by about $K{=}200$ patterns as shown in Figure 1a. Figure 1b illustrates eight of these streamflow patterns, where the first two patterns capture the high flow and subsequent dissipation observed during flood events.

Over the past decade, significant progress has been made in spatiotemporal machine learning (Shi & Yeung, 2018; Wang et al., 2020; Bai et al., 2020). Most existing methods rely on a combination of temporal and spatial encodings to capture interactions among system components, but they often lack interpretability, failing to reveal how forecasts are generated. FDN addresses this limitation with a novel approach that decomposes spatiotemporal systems into a finite set of patterns and then

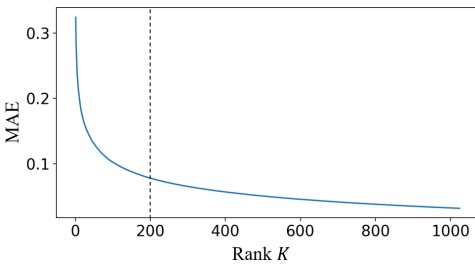 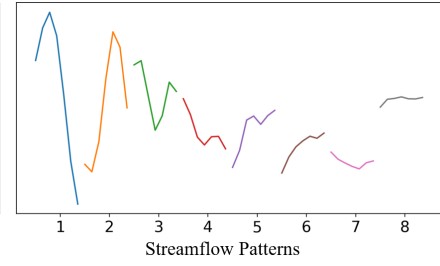

(a) Error in low-rank approximation of $\mathbb{F}$.

(b) Eight most significant patterns using singular value decomposition (SVD) of $\mathbb{F}$.

Figure 1: Low-rank approximation error and important patterns of the 7-day matrix $\mathbb{F} \in \mathbb{R}^{8932 \times 25189}$ of Wabash River's training set. The entire training set ($\mathbb{F}$) can be reasonably approximated by a relatively small ($K = 200$) set of patterns.

uses these patterns for prediction. As a result, FDN delivers accurate forecasts and provides valuable insights into the fundamental patterns driving system behaviors.

The contributions of this work include:

- A novel forecast model architecture utilizing classification and interpolation for direct interpretability.
- The Future Decomposition layer – a novel forecast operator capable of revealing fundamental activity patterns of the system.
- A novel attention layer for localized filtering in multi-variate spatiotemporal systems.
- Using streamflow, traffic, and energy systems, we demonstrate that FDN outperforms state-of-the-art (SOTA) models while providing interpretable forecasts.

## 2 RELATED WORK

Spatiotemporal system forecasting is a highly active sub-field of ML research, primarily originating in the study of traffic systems (Li et al., 2017) and now advancing into multi-domain application (Wu et al., 2020; Cao et al., 2020; Zhou et al., 2021; 2022; Zeng et al., 2023; Majeske & Azad, 2024). In the pursuit of greater forecast accuracy, increasingly sophisticated encoding and decoding schemes have emerged but development of the *forecast operator* has been limited. The forecast operator refers to the inflection point in each forecast model where the past/input sequence is transformed into the future/output sequence. At a high level, contemporary forecast models follow a three-part architecture (shown in Figure 2a) consisting of (1) an encoder module to project the input sequence from input to embedding space (2) a forecast operator to transform the input sequence into an output sequence and (3) a decoder module to project the output sequence from embedding to output space. We summarize recent encoder modules and forecast operators currently in use but we do not cover decoder modules since most methods apply simple linear projection or decoding coincides with the forecast operator (e.g. with $1 \times 1$ kernels in convolution operators, multi-head attention, etc.).

### 2.1 ENCODING MODULES

The encoding module aims to capture information relevant to each node during the projection from input to embedding space. Spatiotemporal systems exhibit both spatial and temporal dynamics which must be properly embedded to support the subsequent forecast operator and decoding module. The system's dependency structure (e.g. streamflow network, road network, etc.) significantly influences the local dynamics of each node, and many methods leverage graph convolution (Kipf & Welling, 2016) to encode it. STGCN, DCRNN, T-GCN, A3T-GCN, and STGM (Yu et al., 2017; Li et al., 2017; Zhao et al., 2019; Bai et al., 2021; Lablack & Shen, 2023) utilize pre-defined graphs but recent methods have opted to learn the dependency structure including MTGNN, StemGNN, AGCRN, SCINet, and MMR-GNN (Wu et al., 2020; Cao et al., 2020; Bai et al., 2020; Liu et al., 2022; Majeske & Azad, 2024).

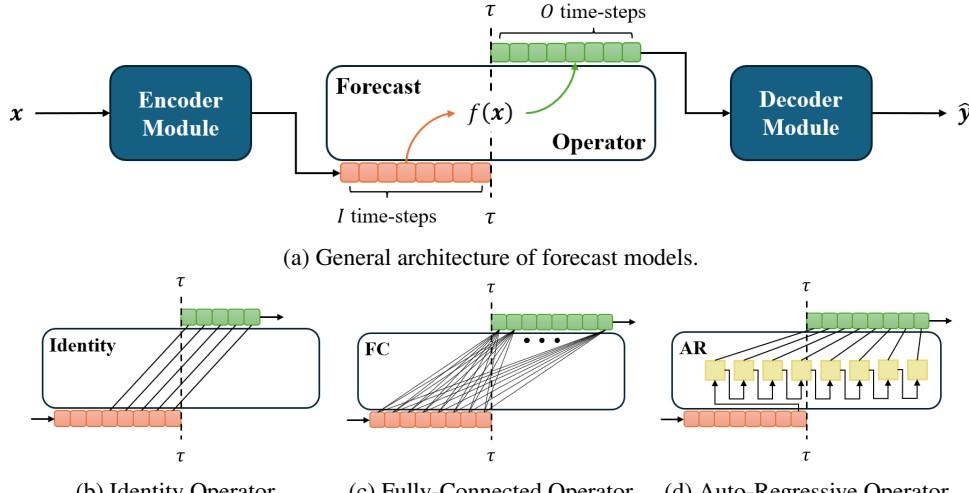

(a) General architecture of forecast models.

(b) Identity Operator     (c) Fully-Connected Operator     (d) Auto-Regressive Operator

Figure 2: Overview of the forecast model architecture and three of the five forecast operators found in recent literature. $\tau$ denotes the current time-step.

Embedding of temporal dynamics continues to develop though many methods still employ RNNs despite their age. While T-GCN, A3T-GCN, and StemGNN (Zhao et al., 2019; Bai et al., 2021; Cao et al., 2020) use vanilla RNN, GRU, or LSTM cells to succeed, other methods (Kratzert et al., 2019; Bai et al., 2020; Majeske & Azad, 2024) have adapted these cells specifically to spatiotemporal data. Temporal convolution networks (TCNs) were introduced in (Lea et al., 2017) and have subsequently been applied in many methods including STGCN, Graph WaveNet, and MTGNN (Yu et al., 2017; Wu et al., 2019; 2020). Recent efforts have adapted the Transformer architecture (Vaswani, 2017) including GMAN and STGM (Zheng et al., 2020; Lablack & Shen, 2023) for traffic forecasting and Informer, Autoformer, and FEDformer (Zheng et al., 2020; Zhou et al., 2021; Wu et al., 2021; Zhou et al., 2022) for general long-term forecasting. New methods continue to arise with SCINet proposing recursive time-series down-sampling and (Zeng et al., 2023) questioning the suitability of Transformer-based forecast models by showing success with simple fully-connected networks.

## 2.2 FORECAST OPERATORS

From the recent literature, we find five forecast operators in use including the identity, fully-connected (FC), convolution, auto-regressive (AR), and attention operators. Section A.2 discusses these operators in detail and table 3 enumerates forecast models that apply them, but we offer a brief description here. The identity operator (Figure 2b) involves selecting the last $O$ elements of the encoded input sequence. The FC operator (Figure 2c) utilizes all-to-all connections to transform the encoded sequence into the output sequence. Convolution operators treat the encoded sequence as image data to transform $I$ input color channels / time-steps into $O$ output color channels / time-steps using $O$ filters of $I$ kernels. The AR operator (Figure 2d) auto-regressively feeds the encoded sequence for $O$ steps to produce the output sequence; typically via an RNN cell. Finally, attention operators compute each element of the output sequence as an attention-weighted sum of the entire input sequence and are central to transformer-based forecast models.

We note that the FC, convolution, and attention operators are fundamentally similar. In fact, $1{\times}1$ and $1{\times}H$ kernels (where $H$ is the embedding dimension) are common and nearly identical to FC. Furthermore, only attention provides direct interpretability through the examination of final attention scores. Our review reveals that the forecast operator has been overlooked in favor of more sophisticated encoding schemes. With FDN, we propose the Future Decomposition layer: a novel forecast operator based on classification and interpolation that can reveal fundamental activity patterns.

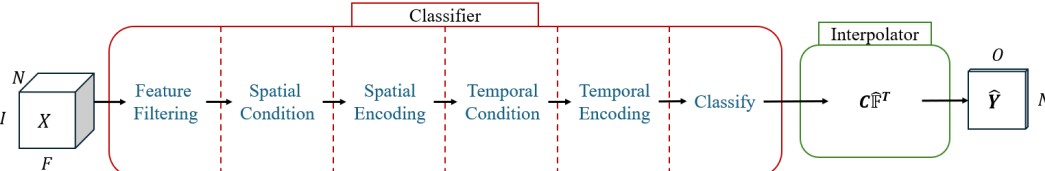

Figure 3: The classifier-interpolator architecture of FDN. Past signals $X$ of each node are soft classified into the likelihood of $K$ possible future patterns. Final prediction $\hat{Y}$ is constructed as an interpolation of the $K$ patterns using the classifier's confidences as weights.

## 3 METHODS

### 3.1 PROBLEM FORMULATION

A spatiotemporal system consists $N$ spatially distributed entities (e.g. solar panels, traffic sensors, stream gauges, etc.) each generating dynamic signals (e.g. power in MW, traffic speed in mph, streamflow in $cm^3$, etc.). The dependency between entities (explicit or correlative) is defined by a graph $G = (V, E)$ with nodes $V$ (entities) and edges $E$ (dependencies). At current time-step $\tau$, each node of the system generates $F$ features (e.g. precipitation, temperature, and streamflow) as $X_\tau \in \mathbb{R}^{N \times F}$ leading to $X \in \mathbb{R}^{N \times T \times F}$ as a sample of $T$ contiguous time-steps. One feature is selected as the forecast target $Y \in X$ and we solve Eq. 1:

$$\arg\min_{\theta} L(Y_{(\tau+1):(\tau+O)}, \mathcal{F}_\theta(X_{(\tau-I+1):\tau}; G)) \tag{1}$$

where $\{X_{\tau-I+1}, X_{\tau-I+2}, ..., X_\tau\} = X_{(\tau-I+1):\tau} \in \mathbb{R}^{N \times I \times F}$ is the observation, $\{y_{\tau+1}, y_{\tau+2}, ..., y_{\tau+O}\} = Y_{(\tau+1):(\tau+O)} \in \mathbb{R}^{N \times O}$ is the horizon, and $L$ is forecast loss. We look to learn $\mathcal{F}_\theta$ capable of predicting the next $O$ time-steps of the target signal $Y_{(\tau+1):(\tau+O)}$ given the past $I$ time-steps of the system $X_{(\tau-I+1):\tau}$ and its dependency structure $G$.

### 3.2 MODEL DESIGN

#### 3.2.1 HIGH-LEVEL ARCHITECTURE

The goal of FDN is to learn patterns of the past (i.e. preambles) that predict particular patterns of the future. To capture the coupling of such past and future patterns, we utilize a classifier-interpolator architecture shown in Figure 3. The classifier aims to determine the correct future pattern based on past activity and features five internal stages to support soft classification accuracy. These include (a) feature filtering to remove noise (b) adding information to identify the location (c) encoding of spatial/dependency dynamics (d) adding information to identify the point-in-time and (e) encoding of temporal dynamics. Future patterns seldom follow past patterns exactly, thus, FDN utilizes soft classification in the selection of a future. Rather than discretize probabilities to select the future pattern of highest likelihood (classification), we apply these probabilities directly (soft classification) to select the future as an interpolation between $K$ futures patterns.

The forward pass of FDN is defined by Eq. 2 where the past of each node is soft classified to produce the $K$-class likelihood matrix $\mathcal{C} \in \mathbb{R}^{N \times K}$. Here, each row vector indicates the classifier's confidence as to which pattern should follow amongst $K$ possibilities. The interpolation module then generates the prediction as a linear combination/interpolation of $K$ patterns using the classifier's confidence scores as weights. The following sections give a detailed discussion of FDN's classifier module and our novel FD layer for prediction and pattern discovery.

$$\text{Classifier}(X_{(\tau-I+1):\tau}, G) \rightarrow \mathcal{C} \in \mathbb{R}^{N \times K}$$
$$\text{Interpolator}(\mathcal{C}, \hat{\mathbb{F}}) \rightarrow \hat{Y}_{(\tau+1):(\tau+O)} \in \mathbb{R}^{N \times O} \tag{2}$$

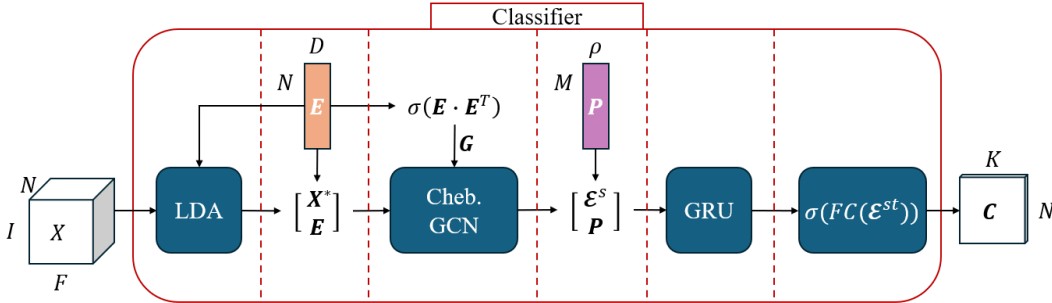

Figure 4: Overview of FDN's classifier module. Features are first filtered by the localized dynamic attention (LDA) layer. Node embeddings are then concatenated onto filtered features for spatial conditioning. Node dependency is then encoded via a Chebyshev GCN layer using the dense graph created from learned node embeddings $\boldsymbol{E}$. Periodic embeddings $\boldsymbol{P}_{(\tau-I+1):\tau}$ are then concatenated for temporal conditioning. A GRU layer encodes the observation window and a fully-connected (FC) layer with softmax activation (denoted by $\sigma$) computes the $K$-class likelihood matrix $\boldsymbol{\mathcal{C}}$.

### 3.2.2 PREAMBLE CLASSIFICATION

FDN's classifier is defined by Eq. 3 and shown in Figure 4 with an accompanying step-by-step description. In effect, this classifier produces a spatiotemporal embedding $\boldsymbol{\mathcal{E}}^{st}$ containing information of each node's past features (in $X_{(\tau-I+1):\tau}$), the past features of its depended nodes (from GCN), features to identify that node and its unique dynamics (from $\boldsymbol{E}$), and features to identity the current moment in time (from $\boldsymbol{P}_{(\tau-I+1):\tau}$). The purpose of LDA, graph convolution, node embeddings, periodic embeddings, and GRU is to encode all relevant information into $\boldsymbol{\mathcal{E}}^{st}$ to maximize preamble classification accuracy.

$$
\begin{aligned}
\sigma(\boldsymbol{E} \cdot \boldsymbol{E}^T) &\to G \\
\text{LDA}(X_{(\tau-I+1):\tau}, \boldsymbol{E}) &\to X^*_{(\tau-I+1):\tau} \in \mathbb{R}^{N \times I \times F} \\
[X^*_{(\tau-I+1):\tau}, \boldsymbol{E}] &\to X^*_{(\tau-I+1):\tau} \in \mathbb{R}^{N \times I \times (F+D)} \\
\text{Chebyshev-GCN}(X^*_{(\tau-I+1):\tau}, G) &\to \mathcal{E}^s \in \mathbb{R}^{N \times I \times H} \\
[\mathcal{E}^s, \boldsymbol{P}_{(\tau-I+1):\tau}] &\to \mathcal{E}^s \in \mathbb{R}^{N \times I \times (H+\rho)} \\
\text{GRU}(\mathcal{E}^s) &\to \boldsymbol{\mathcal{E}}^{st} \in \mathbb{R}^{N \times H} \\
\sigma(\text{FC}(\boldsymbol{\mathcal{E}}^{st})) &\to \boldsymbol{\mathcal{C}} \in \mathbb{R}^{N \times K}
\end{aligned} \tag{3}
$$

### 3.2.3 LEARNED EMBEDDINGS FOR CONDITIONING

To support preamble classification, we spatially and temporally condition the model via two embedding forms learned through the minimization of forecast loss. By conditioning, we refer to the addition of information that identifies the current node (spatial) and point-in-time (temporal) for more precise classification. For spatial conditioning, FDN utilizes learned node embeddings $\boldsymbol{E} \in \mathbb{R}^{N \times D}$ to represent each node's latent dynamics. These node embeddings serve three functions (a) to learn node feature importance and dynamically filter input signals via LDA (b) to add node information (via concatenation onto $X^*_{(\tau-I+1):\tau}$) for spatial conditioning and (c) to learn graph $G$ and encode node inter-dependency via GCN.

To temporally condition the encoding, FDN utilizes learned periodic embeddings $\boldsymbol{P} \in \mathbb{R}^{M \times \rho}$ where $M$ defines the number of moments in a known seasonal period and $\rho$ is embedding dimension. The periodic index function $p(t)$ maps each time-step (as a unique time-stamp) to its moment index $m$ and we apply it at the observation window to retrieve $\boldsymbol{P}_{(\tau-I+1):\tau} \in \mathbb{R}^{I \times \rho}$. These embeddings are then concatenated onto $\mathcal{E}^s$ to condition the encoding to the current moment of the seasonal period. Section A.7 provides evidence of the seasonal period of each forecast variable studied in this paper.

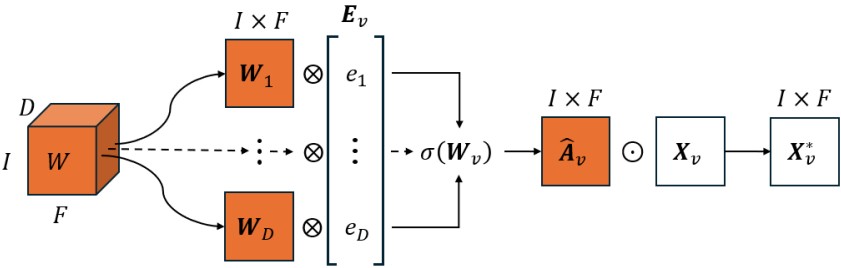

Figure 5: The forward pass of Localized Dynamic Attention on node $v$. Weight matrix $\boldsymbol{W}_v$ is computed as a weighted combination of the $D$ channels in $W$ using node embedding vector $\boldsymbol{E}_v \in \mathbb{R}^D$ as weights. Softmax activation $\sigma(\cdot)$ derives dynamic attention matrix $\hat{\boldsymbol{A}}_v$ and the Hadamard product produces filtered features $\boldsymbol{X}_v^*$.

$$p(t) \rightarrow m \in [1, M]$$
$$\{\boldsymbol{P}_{p(\tau-I+1)}, \boldsymbol{P}_{p(\tau-I+2)}, ..., \boldsymbol{P}_{p(\tau)}\} \rightarrow \boldsymbol{P}_{(\tau-I+1):\tau} \in \mathbb{R}^{I \times \rho} \tag{4}$$

### 3.2.4 LOCALIZED DYNAMIC ATTENTION

In multi-variate settings ($F > 1$), each node generates multiple signals which potentially correlate to the target time series. For example, we should expect traffic volume to have a strong negative correlation with traffic speed (e.g. as volume increases, speed decreases due to congestion). However, these correlations may be highly specific to each node (i.e. localized). For example, the correlation between traffic volume and speed is likely stronger in highways susceptible to congestion (e.g. containing bottlenecks) than in highways that are not. We look to learn these dynamics and accordingly filter node features with LDA.

The forward pass is defined in Eq. 5 where $\hat{A}$ aims to capture the complete attention tensor $A \in \mathbb{R}^{N \times I \times F}$ which defines the exact feature importance at each node. The process to filter the features of node $v$ is demonstrated in figure 5. Specifically, $W$ defines $D$ dynamic attention weight matrices ($W_d \in \mathbb{R}^{I \times F}$) and $\boldsymbol{E}$ defines the mixture of these matrices for each node of the system. By constraining LDA to a lower dimension ($D \ll N$) we can control its precision to avoid over-fitting and reduce memory consumption.

$$W \in \mathbb{R}^{D \times I \times F}$$
$$\sigma(\boldsymbol{E}W) \rightarrow \hat{A} \in \mathbb{R}^{N \times I \times F} \tag{5}$$
$$\hat{A} \odot X_{(\tau-I+1):\tau} \rightarrow X_{(\tau-I+1):\tau}^* \in \mathbb{R}^{N \times I \times F}$$

### 3.2.5 FUTURE DECOMPOSITION LAYER

The classifier produces matrix $\boldsymbol{C} \in \mathbb{R}^{N \times K}$ indicating its confidence of the future pattern amongst $K$ possibilities. For each node, FD uses its likelihood vector to compute the prediction as a linear combination/interpolation of the $K$ patterns. For example, if the classifier shows high confidence of future flooding the prediction will be a unique flood pattern largely constructed from the subset of patterns indicating a flood event. The forward pass is defined by Eq. 6 where $\hat{\mathbb{F}} \in \mathbb{R}^{O \times K}$ is the set of patterns intended to capture $\mathbb{F} \in \mathbb{R}^{N \cdot O \times B}$; the matrix containing all $O$-time-step samples of the system's training set. But, how do we determine $\hat{\mathbb{F}}$?

$$\boldsymbol{C}\hat{\mathbb{F}}^T \rightarrow \hat{\boldsymbol{Y}}_{(\tau+1):(\tau+O)} \in \mathbb{R}^{N \times O} \tag{6}$$

We may apply SVD and take the first $K$ columns of the left-singular matrix $U$ to produce $K$ patterns. However, SVD is cumbersome for sufficiently large systems and we are uncertain of its optimality for forecasting. With this in mind, we design FD to operate on a learned $\hat{\mathbb{F}}$ to automatically discover

Table 1: Technical details of all studied datasets.

| Dataset | Time-steps | Nodes | $F$ | $G$ | Resolution | Horizons |
|---------|-----------|-------|-----|-----|-----------|----------|
| Wabash River | 31,046 | 1,276 | 5 | ✓ | 1 day | 1, 4, 7 |
| E-PEMS-BAY | 52,116 | 325 | 5 | ✓ | 5 minute | 1, 6, 12 |
| Solar-Energy | 52,560 | 137 | 1 | ✗ | 10 minute | 1, 6, 12 |

the $K$ patterns through stochastic gradient descent (SGD). In this way, we can avoid a costly pre-processing step and be certain of the optimality of $\hat{\mathbb{F}}$ to forecasting.

## 4 EXPERIMENTS

We evaluate FDN and all baseline models on three publicly available datasets from hydrology, traffic, and energy. These datasets are formally known as Wabash River, E-PEMS-BAY, and Solar-Energy and we discuss each in detail in Section A.1 of the appendix. Dataset properties are provided in Table 1 including sample size (time-steps), system size (nodes), number of features ($F$), whether a pre-defined graph exists ($G$), sample resolution, and the various prediction horizons we study.

**Data Preparation.** In all experiments, we standardize the features of a node using the mean and standard deviation computed from the training set of that node. All models are trained on standardized features but forecasts are inversely standardized before final evaluation. Only E-PEMS-BAY contains missing values ($\approx 2.5\%$) and we impute with local periodic means. That is, we compute and utilize the periodic mean (separate mean for the 288, 5-minute moments in a day) of each feature and node during imputation.

**Evaluation Metrics.** All models are evaluated according to Mean Absolute Error (MAE), Mean Absolute Percentage Error (MAPE), and Root Mean Square Error (RMSE). Metrics are masked to exclude imputed values and ensure model performance quantification is a consequence of forecasts made on ground truth samples only. Imputed values are also masked in the computation of forecast loss during SGD. Experiments are executed three times using three pseudo-randomly generated initialization seeds and results are given as the mean and standard deviation of these trials. Due to space limitations, standard deviations are presented in section A.3 of the appendix.

**Model Baselines.** We evaluate FDN against 11 forecast models found throughout the literature. This includes simpler models designed for single time-series forecasting, complex models designed for multiple time-series, and highly sophisticated SOTA models designed for multiple time-series forecasting across multiple domains. Implementations of these models were acquired from their published GitHub repositories except for T-GCN and A3T-GCN implemented in PyTorch Geometric Temporal (Rozemberczki et al., 2021). All experiments were conducted on an Nvidia A100 GPU with 40GB of memory. Each model is trained using MAE (PyTorch's L1Loss) as forecast loss.

### 4.1 MAIN RESULTS

Forecast performance metrics for all models and prediction horizons are presented in Table 2. Model efficiency metrics are reported and discussed at length in section A.4 due to space limitations. Overall, FDN matches or exceeds the performance of other SOTA methods. For longer prediction horizons, FDN consistently outperforms the next best model across all datasets. The most notable improvement is observed for E-PEMS-BAY, with a 9.1% reduction in MAPE and a 2.5% reduction in RMSE. FDN also gives a 6.3% MAPE reduction while nearly matching RMSE for Wabash River, and a 1% and 12% reduction in MAPE and RMSE for Solar-Energy for the largest horizon.

Since the metrics of Table 2 represent mean forecast performance over hundreds of nodes, it is difficult to gauge improvement fully. Figures 9, 10, and 11 from section A.6 plot percentage change in MAPE and RMSE (where negative indicates improvement) at all nodes of Wabash River, E-PEMS-BAY, and Solar-Energy for FDN relative to the second-best model. The x-axis shows baseline model performance while node color intensity / size is determined by the coefficient of variation (CoV) in the forecast variable. Here, we observe greater improvement at higher variance nodes suggesting FDN is particularly suited to capturing large changes in the forecast variable.

Table 2: Average MAE, MAPE, and RMSE from all horizons on three datasets. The best performance is emboldened while the second-best is underlined. GCN-based models, which require a pre-defined graph, are incompatible with Solar-Energy data, as it lacks such a structure. "N/A" indicates this model incompatibility. The standard deviations are presented in the appendix.

| | Horizon | Metric | GRU | TCN | FED-former | LTSF DLinear | T-GCN | A3T-GCN | STGM | Stem-GNN | MTGNN | AGCRN | SCINet | FDN |
|---|---|---|---|---|---|---|---|---|---|---|---|---|---|---|
| **Wabash River** | 1 | MAE | 3.517 | 3.499 | 3.868 | 3.631 | 6.975 | 7.035 | 3.938 | 3.700 | 3.355 | **2.945** | 3.476 | _3.127_ |
| | | MAPE | _17.210_ | 17.731 | 31.746 | 17.508 | 26.657 | 26.684 | 25.480 | 19.087 | 19.943 | 17.658 | 18.367 | **16.292** |
| | | RMSE | 10.655 | 10.678 | 10.068 | 10.972 | 14.658 | 14.788 | 10.963 | 11.036 | 10.015 | **8.978** | 10.425 | _9.583_ |
| | 4 | MAE | 7.173 | 7.400 | 7.634 | 7.326 | 9.664 | 9.696 | 8.515 | 7.363 | 6.895 | _6.649_ | 7.444 | **6.624** |
| | | MAPE | 28.337 | 28.542 | 41.893 | 28.166 | 35.477 | 35.684 | 43.242 | 29.565 | 30.719 | _27.996_ | 31.434 | **26.183** |
| | | RMSE | 18.727 | 19.306 | 18.368 | 19.125 | 21.067 | 21.086 | 19.563 | 19.037 | 18.124 | **17.518** | 18.985 | _17.890_ |
| | 7 | MAE | 9.604 | 9.812 | 10.153 | 9.887 | 11.552 | 11.575 | 11.028 | 10.025 | 9.242 | _9.128_ | 10.184 | **9.122** |
| | | MAPE | 35.441 | 39.157 | 47.402 | _35.063_ | 41.771 | 41.574 | 47.421 | 39.401 | 36.663 | 35.085 | 40.243 | **32.978** |
| | | RMSE | 23.175 | 23.742 | 23.022 | 23.803 | 24.882 | 24.898 | 24.170 | 23.824 | **22.473** | 22.719 | 23.604 | _22.611_ |
| **E-PEMS-BAY** | 1 | MAE | 0.974 | 0.942 | 1.354 | 1.000 | 2.040 | 2.021 | 1.558 | **0.916** | 1.185 | 1.034 | 0.987 | _0.937_ |
| | | MAPE | 1.912 | _1.835_ | 2.797 | 1.945 | 4.207 | 4.157 | 2.909 | **1.787** | 2.707 | 2.046 | 1.962 | 1.875 |
| | | RMSE | 1.826 | 1.776 | 2.492 | 1.987 | 3.280 | 3.284 | 2.398 | **1.746** | 2.199 | 1.868 | 1.866 | _1.757_ |
| | 6 | MAE | 1.587 | 1.588 | 2.021 | 1.689 | 2.420 | 2.383 | 5.380 | _1.549_ | 1.817 | 1.705 | 1.598 | **1.475** |
| | | MAPE | 3.334 | 3.279 | 4.232 | 3.493 | 5.109 | 5.005 | 10.242 | _3.263_ | 4.064 | 3.571 | 3.402 | **3.106** |
| | | RMSE | 3.414 | 3.502 | 3.974 | 3.664 | 4.200 | 4.218 | 7.345 | 3.276 | 3.749 | 3.406 | _3.181_ | **3.076** |
| | 12 | MAE | 2.076 | 2.117 | 2.329 | 2.225 | 2.708 | 2.719 | 12.531 | 1.971 | 2.604 | 2.083 | _1.950_ | **1.792** |
| | | MAPE | 4.459 | 4.555 | 4.891 | 4.684 | 5.781 | 5.774 | 21.043 | 4.245 | 5.929 | 4.371 | _4.208_ | **3.857** |
| | | RMSE | 4.465 | 4.655 | 4.657 | 4.858 | 4.900 | 4.955 | 16.393 | 4.125 | 5.364 | 4.166 | _3.897_ | **3.803** |
| **Solar-Energy** | 1 | MAE | 0.292 | 0.293 | 0.915 | 0.319 | N/A | N/A | N/A | 1.419 | 0.285 | _0.247_ | 0.277 | **0.238** |
| | | MAPE | 68.163 | 67.696 | 71.959 | 68.519 | N/A | N/A | N/A | 76.452 | 68.097 | _66.927_ | 67.037 | 66.815 |
| | | RMSE | 0.817 | 0.786 | 1.600 | 0.854 | N/A | N/A | N/A | 2.840 | 0.779 | **0.704** | 0.768 | _0.705_ |
| | 6 | MAE | 0.752 | 0.789 | 1.281 | 0.987 | N/A | N/A | N/A | 2.032 | _0.590_ | 0.592 | 0.642 | **0.557** |
| | | MAPE | 73.144 | 73.410 | 74.525 | 74.566 | N/A | N/A | N/A | 79.889 | 71.429 | _71.249_ | 71.954 | **70.899** |
| | | RMSE | 1.894 | 1.902 | 2.244 | 2.249 | N/A | N/A | N/A | 3.961 | _1.511_ | 1.518 | 1.583 | **1.422** |
| | 12 | MAE | 1.241 | 1.451 | 1.788 | 1.645 | N/A | N/A | N/A | 2.092 | _0.870_ | 0.901 | 0.962 | **0.869** |
| | | MAPE | 76.957 | 77.961 | 76.697 | 78.337 | N/A | N/A | N/A | 80.719 | _74.019_ | 74.049 | 74.670 | **73.267** |
| | | RMSE | 2.950 | 3.166 | 3.027 | 3.472 | N/A | N/A | N/A | 4.117 | _2.215_ | 2.270 | 2.332 | **1.977** |

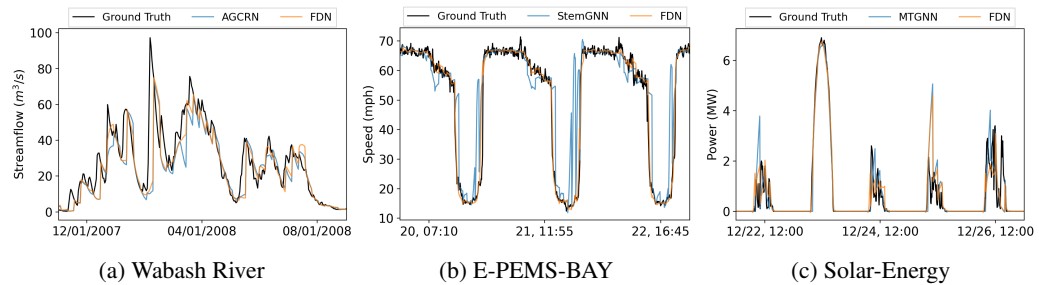

(a) Wabash River      (b) E-PEMS-BAY      (c) Solar-Energy

Figure 6: Forecasts of select nodes in Wabash River, E-PEMS-BAY, and Solar-Energy. The ground truth signal is shown in black, the second-best model in blue, and FDN in orange.

Predictions from FDN and the second-best performer are shown in Figure 6. FDN shows an improvement during high streamflow events in the Wabash River by capturing the many peaks more closely than AGCRN. In Solar-Energy, FDN shows less over-prediction relative to MTGNN during the peak hours of early afternoon. Finally, in E-PEMS-BAY, FDN predicts the sudden halt of traffic during rush hour and returned flow in the late evenings whereas StemGNN is late and early to predict these events. FDN delivers accurate forecasts and, as we will see in the next section, its classifier-interpolator architecture allows us to easily interpret its prediction process.

## 4.2 LEARNED PATTERNS AND INTERPRETABILITY

Figure 7 shows FDN's prediction process for streamflow, traffic speed, and power production in the left, middle, and right columns respectively. The top row shows a real-time prediction where dashed vertical lines indicate the observation and horizon windows. The observed/preamble signal, shown in the observation window as green, is classified to produce the next forecast, shown in the horizon window as dotted red. The preamble classification is used to interpolate from FD's $K$ learned

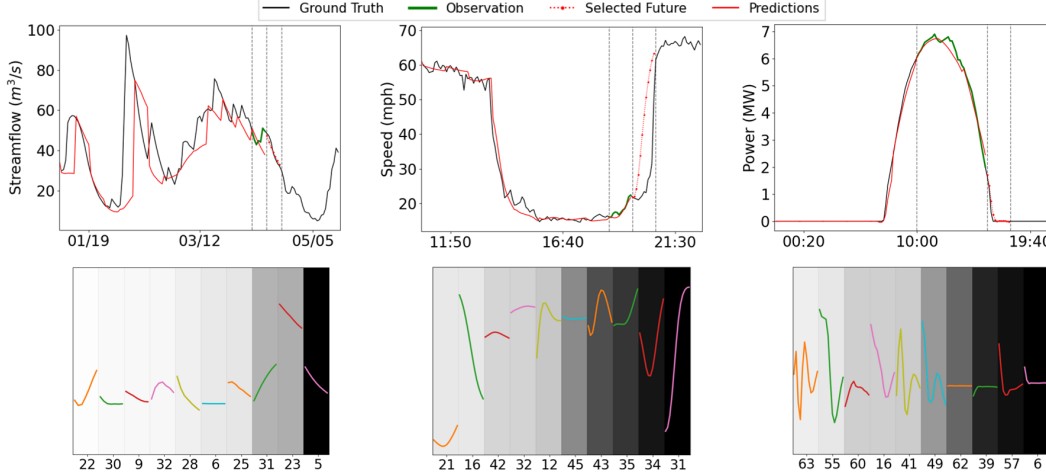

Figure 7: Real-time FDN predictions (top row) for Wabash River, E-PEMS-BAY, and Solar-Energy, in left, middle, and right columns respectively. We show ground truth in black, past predictions in red, observation/preamble in green, and selected/interpolated patterns for the next forecast in dashed red lines. In the bottom row, we show ten of the learned patterns and indicate current soft classification probability by the darkness of their background. That is, the patterns are arranged from left to right in the ascending order of the classifier's confidence.

patterns, shown in the bottom row. For clarity, we show the top ten patterns, with their respective likelihood indicated by the darkness of their background.

Considering the figures more closely, FDN's predictions become clear. On the descent from a period of high streamflow, FDN predicts this process to continue with high confidence in a "descent pattern". Towards the end of a period of traffic congestion, we can see FDN has detected the uptick in vehicle speed and correctly predicts the return of traffic flow. Finally, FDN correctly detects the halt of solar power production at approximately 4:30pm; the sunset time for Alabama, USA in late December 2006.

Overall, FDN shows remarkable interpretability. Through classification, we can directly observe the choice of FDN's next forecast. Moreover, learned patterns reveal some of the fundamental activities present in each system. Here we observe a few patterns that indicate flood dissipation, traffic relief, and time of sunset.

### 4.3 MODEL GENERALIZATION

We can think of the FD layer as attempting to learn $K$ vectors which capture all information of the training set $\mathbb{F} \in \mathbb{R}^{N \cdot O \times B}$, similar to SVD. Noting that $\hat{\mathbb{F}}$ is a set of vectors in $O$ dimensions, FD attempts to learn a vector space that encloses $\mathbb{F}$ in its entirety. Figure 8a shows the 64 learned patterns (in black) and all ground truth and predicted samples for E-PEMS-BAY reduced to two dimensions by principle component analysis (PCA). Black dashed lines connect the outer-most learned patterns as a convex hull to indicate FD's learned vector space. In this case, we see evidence of good generalization as $\hat{\mathbb{F}}$ nearly captures all training set samples, shown in blue, and all testing set samples, shown in orange. Additionally, the high level of coverage of testing set samples (orange) by testing set predictions (red) indicates forecast accuracy.

### 4.4 ABLATION STUDY

We now study the contribution of each component in FDN to forecast accuracy. Specifically, we test (a) increasing the number of learned patterns (b) no attention versus various attention layers including LDA (c) node inter-dependency learning from GCN (d) learned node embedding dimension (e) regularization of the learned patterns (f) moment resolution and (g) learned periodic embedding dimension. Each ablation study was conducted on the longest prediction horizon and results show the

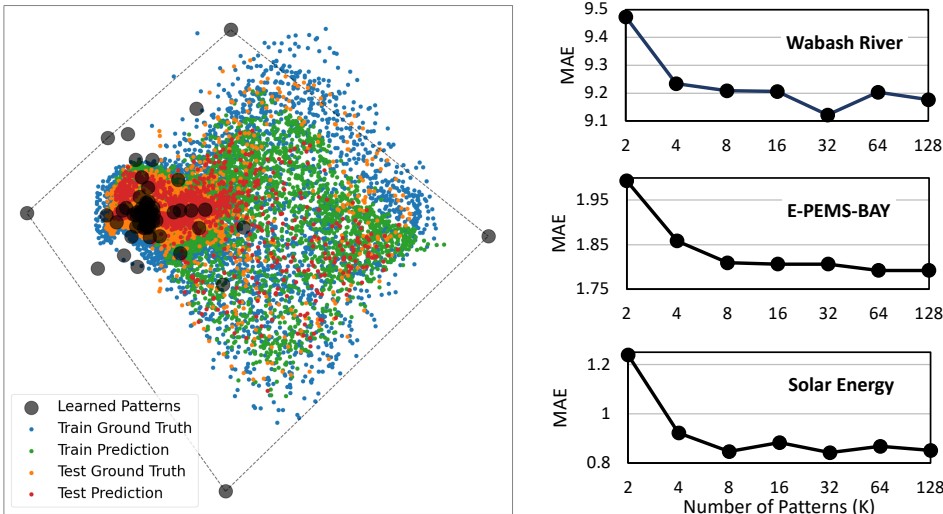

(a) FDN's fit to training (blue) and testing (orange) sets, along with the train (green) and test (red) predictions, within the vector space learned by FD's $K$ patterns.

(b) Ablation study results for the number of learned patterns $K$.

Figure 8: (a) Model generalization by the learned patterns and (b) the impact $K$ on forecasting error.

average of three trials. We discuss the first ablation below showing results in figure 8b but discuss the other studies and their results in section A.5 due to space limitations.

**Learned Patterns.** The FD layer learns a set of patterns to capture $\mathbb{F}$ which, as demonstrated in Figure 1a, is greatly benefited by increasing the rank / number of patterns. Here, we test increasing the number of patterns learned by FD to improve its ability to capture $\mathbb{F}$. Table 10 and Figure 8b demonstrate the effectiveness of FD as we observe a saturation in forecast performance when learning as few as eight patterns.

**Node Embedding Dimension.** FDN's learned node embeddings condition the classifier to each node, determine their feature filtering, and learn the graph for dependency encoding. The node embedding dimension controls the specificity of node conditioning, LDA, and the learned graph and must be tuned for proper generalization. Table 13 shows the result of increasing dimension $D$ and we observe a saturation in forecasting performance at approximately $D = 10$.

**Periodic Moment Resolution.** Periodic embeddings consist of a sequence of $\rho$-dimensional embeddings representing moments in the known period/season of the forecast variable. Moment resolution ranges from the duration of the period ($M = 1$) to the duration of each time step ($M \gg 1$) and must be tuned to avoid over-fitting. Table 15 shows increasing moment resolution starting from period/season duration ($M = 1$) and increasing up to time-step duration ($M = 366$ in Wabash River). Wabash River shows saturation at 3-months (capturing the 4 seasons of the year) while E-PEMS-BAY and Solar-Energy benefit from high-resolution moments.

## 5 CONCLUSION

This paper presents FDN, a novel forecast model architecture which leverages classification and interpolation to produce accurate and interpretable forecasts. FDN utilizes the Future Decomposition layer, a new forecast operator to the literature capable of revealing latent patterns of the target time-series. We demonstrate FDN's forecast accuracy by meeting or exceeding the performance of current SOTA forecast models across three datasets from hydrologic, traffic, and energy systems. Finally, FDN shows exceptional efficiency with faster epoch runtimes and far fewer parameters than its competitors. We are excited to present FDN and feel confident its novel architecture can inspire new avenues of spatiotemporal forecasting research that advance interpretability.

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

# A APPENDIX

## A.1 STUDIED DATASETS

This section offers a detailed description of the three datasets used to evaluate FDN: Wabash River, E-PEMS-BAY, and Solar-Energy.

**Wabash River.** The Wabash River dataset (Majeske et al., 2022) contains many hydrologic and meteorologic features recorded at various gauging stations across the Wabash River Basin. This basin spans three US states including eastern Illinois, western Ohio, and central Indiana and consists 1276 subbasins (nodes). Measurements of temperature (min and max), precipitation, soil water, and

streamflow are recorded at each subbasin in 1-day intervals. For each subbasin, we utilize the past seven days of all five features to forecast streamflow for the next one, four, and seven days. The Wabash River dataset contains a pre-defined dependency structure in the form of its streamflow network (a tree).

**E-PEMS-BAY.** The E-PEMS-BAY dataset contains many highway traffic features recorded from a sample of the Caltrans PeMS's (Varaiya, 2007) traffic sensor network. Specifically, this dataset contains samples drawn from 325 sensors (nodes) of the north-western region of California's Santa Clara district. These sensors record *total samples* (across all lanes), *percent observed* (non-imputed data points), *total flow* (vehicles/5-min), *average occupancy* (as a 0-1 rate), and *average speed* (mph) in 5-minute intervals. For each sensor, we consider the past hour (12, 5-minute time-steps) of all five features to forecast *average speed* for the next 5, 30, and 60 minutes (1, 6, and 12 time-steps). E-PEMS-BAY includes a pre-defined dependency structure but it is inferred from sensor features (Majeske & Azad, 2024) rather than a ground truth network.

**Solar-Energy.** The Solar-Energy dataset contains synthetic solar photovoltaic power plant samples produced by a 2006 integration study (NREL, 2006) of the US. In this work, we consider the 137 solar power plants from Alabama state following (Lai et al., 2018; Wu et al., 2020; Liu et al., 2022). Only one feature is recorded at each plant (node) of this system which is the photovoltaic power (in mega-Watts) produced. The original dataset comes in 5-minute resolution but we use the down-sampled (10-minute) version following many others (Lai et al., 2018; Wu et al., 2020; Liu et al., 2022). For each plant, we utilize the past six hours (36, 10-minute time-steps) of photovoltaic power to forecast the next 10, 60, and 120 minutes of photovoltaic power (1, 6, and 12 time-steps). No pre-defined dependency structure exists for the 137 power plants.

## A.2 RELATED WORK CONTINUED

This section offers a more detailed discussion of each forecast operator including their core operation and some limitations. We refer to the input/encoded sequence $x$ as containing $I$ time-steps, and output/decoded sequence $\hat{y}$ as containing $O$ time-steps, and current time-step as $\tau$.

**Identity Op.** The identity operator (Figure 2b) involves selecting the last $O$ elements of the encoded input sequence. This operator requires the input sequence be equal to or greater in length than the output sequence. For certain encoders, such as RNNs, important information may be omitted since only the final element of the output sequence is a function of all input time-steps.

**Fully-Connected Op.** The fully-connected (FC) operator (Figure 2c) utilizes all-to-all connections to transform the encoded sequence into the output sequence. As a result, each of the $O$ output time-steps are a function of all $I$ input time-steps. This allows any arbitrary mapping $I \rightarrow O$ but incorporates all input time-steps which may contain redundant/noisy information for long sequences.

**Convolution Op.** This operator applies a convolution layer by treating the encoded sequence as image data where time-steps are handled as color-channels and filters. Specifically, $I$ input time-steps are transformed into $O$ output time-steps by applying a convolution layer of $O$ filters each with $I$ kernels. The operator can perform any mapping $I \rightarrow O$ and is very similar to FC since each output time-step is a summation of kernels applied at every input time-step.

**Auto-Regressive Op.** The auto-regressive (AR) operator (Figure 2d) recurrently feeds the encoded sequence for $O$ steps to produce the output sequence. The AR operator is primarily seen in recurrent neural networks (RNNs) where a decoder cell auto-regressively feeds the encoded sequence produced by a separate encoder cell. AR can perform any mapping $I \rightarrow O$ and output time-steps are strictly causal but RNNs bring challenges to gradient stability.

**Attention Op.** Attention operators compute each element of the output sequence as an attention-weighted sum of the entire input sequence. This operator was popularized by Transformers (Vaswani, 2017) (designed for language translation) but recent methods (Zhou et al., 2021; Wu et al., 2021; Zhou et al., 2022) have adapted the Transformer architecture to long-term series fore-casting. Specifically, these methods zero-pad the latter half of the input sequence to match the output sequence length and use it as the query. The encoded input sequence is used as key and value and fed with the query to multi-head attention to produce the output sequence.

Table 3: Forecast operators of models found throughout recent literature.

| Forecast Operator | Forecast Models |
|---|---|
| Identity | EA-LSTM Kratzert et al. (2019), STGCN Yu et al. (2017) |
| Fully-Connected (FC) | LTSF_Linear, LTSF_NLinear, LTSF_DLinear Zeng et al. (2023) T-GCN Zhao et al. (2019), A3T-GCN Bai et al. (2021), StemGNN Cao et al. (2020) |
| Convolution | ASTGCN Guo et al. (2019), Graph WaveNet Wu et al. (2019) MTGNN Wu et al. (2020), AGCRN Bai et al. (2020) SCINet Liu et al. (2022), STGM Lablack & Shen (2023) |
| Auto-Regression (AR) | RNN, GRU, LSTM Majeske et al. (2022), TCN Lea et al. (2017) MMR-GNN Majeske & Azad (2024) |
| Attention | GMAN Zheng et al. (2020), Informer Zhou et al. (2021) Autoformer Wu et al. (2021), FEDformer Zhou et al. (2022) |
| Classifier-Interpolator (CI) | FDN |

Table 4: Forecast MAE, MAPE, and RMSE from all horizons on the Wabash River Basin.

| Horizon Metric | 1 MAE | MAPE | RMSE | 4 MAE | MAPE | RMSE | 7 MAE | MAPE | RMSE |
|---|---|---|---|---|---|---|---|---|---|
| GRU | 3.517 ± 0.004 | 17.210 ± 0.186 | 10.655 ± 0.060 | 7.173 ± 0.032 | 28.337 ± 0.158 | 18.727 ± 0.052 | 9.604 ± 0.011 | 35.441 ± 0.278 | 23.175 ± 0.018 |
| TCN | 3.499 ± 0.005 | 17.731 ± 0.470 | 10.678 ± 0.019 | 7.400 ± 0.079 | 28.542 ± 0.864 | 19.306 ± 0.218 | 9.812 ± 0.033 | 39.157 ± 0.807 | 23.742 ± 0.096 |
| FEDformer | 3.868 ± 0.002 | 31.746 ± 0.007 | 10.068 ± 0.009 | 7.634 ± 0.017 | 41.893 ± 0.465 | 18.368 ± 0.018 | 10.153 ± 0.006 | 47.402 ± 0.011 | 23.022 ± 0.017 |
| LTSF_DLinear | 3.631 ± 0.000 | 17.508 ± 0.000 | 10.972 ± 0.000 | 7.326 ± 0.000 | 28.166 ± 0.000 | 19.125 ± 0.000 | 9.887 ± 0.000 | 35.063 ± 0.001 | 23.803 ± 0.000 |
| TGCN | 6.975 ± 0.023 | 26.657 ± 0.404 | 14.658 ± 0.015 | 9.664 ± 0.012 | 35.477 ± 0.484 | 21.067 ± 0.038 | 11.552 ± 0.034 | 41.771 ± 0.303 | 24.882 ± 0.110 |
| A3TGCN | 7.035 ± 0.010 | 26.684 ± 0.095 | 14.788 ± 0.010 | 9.696 ± 0.036 | 35.684 ± 0.199 | 21.086 ± 0.075 | 11.575 ± 0.015 | 41.574 ± 0.242 | 24.898 ± 0.013 |
| STGM | 3.938 ± 0.000 | 25.480 ± 0.000 | 10.963 ± 0.000 | 8.515 ± 0.103 | 43.242 ± 0.192 | 19.563 ± 0.132 | 11.028 ± 0.000 | 47.421 ± 0.000 | 24.170 ± 0.000 |
| StemGNN | 3.700 ± 0.057 | 19.087 ± 0.340 | 11.036 ± 0.294 | 7.363 ± 0.020 | 29.565 ± 0.432 | 19.037 ± 0.016 | 10.025 ± 0.172 | 39.401 ± 1.802 | 23.824 ± 0.233 |
| MTGNN | 3.355 ± 0.107 | 19.943 ± 1.140 | 10.015 ± 0.271 | 6.895 ± 0.052 | 30.719 ± 1.019 | 18.124 ± 0.046 | 9.242 ± 0.075 | 36.663 ± 0.353 | **22.473 ± 0.294** |
| AGCRN | **2.945 ± 0.033** | 17.658 ± 0.453 | **8.978 ± 0.143** | **6.649 ± 0.006** | 27.996 ± 0.325 | **17.518 ± 0.068** | 9.128 ± 0.017 | 35.085 ± 0.141 | 22.719 ± 0.082 |
| SCINet | 3.476 ± 0.079 | 18.367 ± 0.026 | 10.425 ± 0.319 | 7.444 ± 0.057 | 31.434 ± 0.406 | 18.985 ± 0.047 | 10.184 ± 0.084 | 40.243 ± 0.922 | 23.604 ± 0.135 |
| FDN | 3.127 ± 0.011 | **16.292 ± 0.108** | 9.583 ± 0.060 | **6.624 ± 0.009** | **26.183 ± 0.599** | 17.890 ± 0.044 | **9.122 ± 0.029** | **32.978 ± 0.070** | 22.611 ± 0.015 |

Table 5: Forecast MAE, MAPE, and RMSE from all horizons on E-PEMS-BAY.

| Horizon Metric | 1 MAE | MAPE | RMSE | 6 MAE | MAPE | RMSE | 12 MAE | MAPE | RMSE |
|---|---|---|---|---|---|---|---|---|---|
| GRU | 0.974 ± 0.013 | 1.912 ± 0.022 | 1.826 ± 0.019 | 1.587 ± 0.012 | 3.334 ± 0.036 | 3.414 ± 0.016 | 2.076 ± 0.002 | 4.459 ± 0.059 | 4.465 ± 0.016 |
| TCN | 0.942 ± 0.019 | 1.835 ± 0.025 | 1.776 ± 0.023 | 1.588 ± 0.002 | 3.279 ± 0.010 | 3.502 ± 0.008 | 2.117 ± 0.006 | 4.555 ± 0.079 | 4.655 ± 0.016 |
| FEDformer | 1.354 ± 0.003 | 2.797 ± 0.006 | 2.492 ± 0.007 | 2.021 ± 0.047 | 4.232 ± 0.087 | 3.974 ± 0.042 | 2.329 ± 0.007 | 4.891 ± 0.013 | 4.657 ± 0.007 |
| LTSF_DLinear | 1.000 ± 0.000 | 1.945 ± 0.000 | 1.987 ± 0.000 | 1.689 ± 0.000 | 3.493 ± 0.000 | 3.664 ± 0.000 | 2.225 ± 0.000 | 4.684 ± 0.000 | 4.858 ± 0.000 |
| TGCN | 2.040 ± 0.030 | 4.207 ± 0.068 | 3.280 ± 0.025 | 2.420 ± 0.021 | 5.109 ± 0.044 | 4.200 ± 0.034 | 2.708 ± 0.016 | 5.781 ± 0.011 | 4.900 ± 0.002 |
| A3TGCN | 2.021 ± 0.044 | 4.157 ± 0.089 | 3.284 ± 0.040 | 2.383 ± 0.015 | 5.005 ± 0.035 | 4.218 ± 0.009 | 2.719 ± 0.018 | 5.774 ± 0.048 | 4.955 ± 0.012 |
| STGM | 1.558 ± 0.225 | 2.909 ± 0.324 | 2.398 ± 0.147 | 5.380 ± 0.186 | 10.242 ± 0.415 | 7.345 ± 0.225 | 12.531 ± 1.749 | 21.043 ± 2.548 | 16.393 ± 2.619 |
| StemGNN | **0.916 ± 0.006** | **1.787 ± 0.015** | **1.746 ± 0.016** | 1.549 ± 0.045 | 3.263 ± 0.079 | 3.276 ± 0.048 | 1.971 ± 0.028 | 4.245 ± 0.040 | 4.125 ± 0.029 |
| MTGNN | 1.185 ± 0.081 | 2.707 ± 0.312 | 2.199 ± 0.130 | 1.817 ± 0.037 | 4.064 ± 0.098 | 3.749 ± 0.046 | 2.604 ± 0.137 | 5.929 ± 0.288 | 5.364 ± 0.280 |
| AGCRN | 1.034 ± 0.014 | 2.046 ± 0.018 | 1.868 ± 0.010 | 1.705 ± 0.009 | 3.571 ± 0.024 | 3.406 ± 0.025 | 2.083 ± 0.078 | 4.371 ± 0.114 | 4.166 ± 0.160 |
| SCINet | 0.987 ± 0.005 | 1.962 ± 0.019 | 1.866 ± 0.003 | 1.598 ± 0.006 | 3.402 ± 0.033 | 3.181 ± 0.015 | 1.950 ± 0.006 | 4.208 ± 0.021 | 3.897 ± 0.020 |
| FDN | 0.937 ± 0.010 | 1.875 ± 0.025 | 1.757 ± 0.017 | **1.475 ± 0.029** | **3.106 ± 0.081** | **3.076 ± 0.043** | **1.792 ± 0.024** | **3.857 ± 0.083** | **3.803 ± 0.031** |

Table 6: Forecast MAE, MAPE, and RMSE from all horizons on Solar-Energy.

| Horizon Metric | 1 MAE | MAPE | RMSE | 6 MAE | MAPE | RMSE | 12 MAE | MAPE | RMSE |
|---|---|---|---|---|---|---|---|---|---|
| GRU | 0.292 ± 0.003 | 68.163 ± 0.025 | 0.817 ± 0.006 | 0.752 ± 0.003 | 73.144 ± 0.073 | 1.894 ± 0.001 | 1.241 ± 0.045 | 76.957 ± 0.309 | 2.950 ± 0.067 |
| TCN | 0.293 ± 0.001 | 67.696 ± 0.117 | 0.786 ± 0.003 | 0.789 ± 0.009 | 73.410 ± 0.139 | 1.902 ± 0.017 | 1.451 ± 0.022 | 77.961 ± 0.282 | 3.166 ± 0.046 |
| FEDformer | 0.915 ± 0.023 | 71.959 ± 0.097 | 1.600 ± 0.023 | 1.281 ± 0.007 | 74.525 ± 0.029 | 2.244 ± 0.001 | 1.788 ± 0.034 | 76.697 ± 0.249 | 3.027 ± 0.035 |
| LTSF_DLinear | 0.319 ± 0.000 | 68.519 ± 0.000 | 0.854 ± 0.000 | 0.987 ± 0.000 | 74.566 ± 0.000 | 2.249 ± 0.000 | 1.645 ± 0.000 | 78.337 ± 0.010 | 3.472 ± 0.000 |
| TGCN | N/A | N/A | N/A | N/A | N/A | N/A | N/A | N/A | N/A |
| A3TGCN | N/A | N/A | N/A | N/A | N/A | N/A | N/A | N/A | N/A |
| STGM | N/A | N/A | N/A | N/A | N/A | N/A | N/A | N/A | N/A |
| StemGNN | 1.419 ± 0.157 | 76.452 ± 0.852 | 2.840 ± 0.452 | 2.032 ± 0.121 | 79.889 ± 0.536 | 3.961 ± 0.252 | 2.092 ± 0.175 | 80.719 ± 0.277 | 4.117 ± 0.391 |
| MTGNN | 0.285 ± 0.010 | 68.097 ± 0.105 | 0.779 ± 0.012 | 0.590 ± 0.006 | 71.430 ± 0.101 | 1.511 ± 0.012 | 0.870 ± 0.018 | 74.019 ± 0.113 | 2.215 ± 0.028 |
| AGCRN | 0.247 ± 0.004 | 66.927 ± 0.060 | **0.704 ± 0.001** | 0.592 ± 0.007 | 71.249 ± 0.094 | 1.518 ± 0.012 | 0.901 ± 0.006 | 74.049 ± 0.080 | 2.270 ± 0.008 |
| SCINet | 0.277 ± 0.005 | 67.037 ± 0.310 | 0.768 ± 0.005 | 0.642 ± 0.004 | 71.954 ± 0.050 | 1.583 ± 0.007 | 0.962 ± 0.007 | 74.670 ± 0.051 | 2.332 ± 0.013 |
| FDN | **0.238 ± 0.001** | **66.815 ± 0.006** | 0.705 ± 0.001 | **0.557 ± 0.005** | **70.899 ± 0.058** | **1.422 ± 0.005** | **0.869 ± 0.021** | **73.267 ± 0.193** | **1.977 ± 0.035** |

## A.3 ADDITIONAL RESULTS

This section includes the extended results for all horizons on each dataset. Tables 4, 5, and 6 provide mean and standard deviation of MAE, MAPE, and RMSE across the three trials for Wabash River, E-PEMS-BAY, and Solar-Energy respectively.

Table 7: Total model parameters and average epoch runtime from all horizons on Wabash River.

| Horizon | 1 | | 4 | | 7 | |
|---|---|---|---|---|---|---|
| Metric | Parameters | Runtime | Parameters | Runtime | Parameters | Runtime |
| GRU | 2753 | $2.030 \pm 0.060$ | 2753 | $2.540 \pm 0.016$ | 2753 | $3.034 \pm 0.042$ |
| TCN | 3025 | $9.821 \pm 0.159$ | 2833 | $35.847 \pm 0.183$ | 2833 | $61.514 \pm 0.136$ |
| FEDformer | 17263884 | $8.769 \pm 0.278$ | 17329420 | $9.010 \pm 0.477$ | 17460492 | $9.252 \pm 0.231$ |
| LTSF_DLinear | 20416 | $259.021 \pm 3.431$ | 81664 | $256.407 \pm 5.117$ | 142912 | $255.115 \pm 3.483$ |
| TGCN | 25985 | $16.810 \pm 0.045$ | 26180 | $16.784 \pm 0.101$ | 26375 | $17.844 \pm 0.082$ |
| A3TGCN | 62208 | $207.513 \pm 0.057$ | 62511 | $207.190 \pm 0.470$ | 62814 | $198.638 \pm 7.634$ |
| STGM | 777065 | $271.4 \pm 0$ | 777065 | $269.820 \pm 0.276$ | 777065 | $272.305 \pm 0$ |
| StemGNN | 5275264 | $95.957 \pm 9.793$ | 5275288 | $94.023 \pm 8.233$ | 5275312 | $99.721 \pm 3.392$ |
| MTGNN | 24676161 | $1650.085 \pm 0.013$ | 1930420 | $88.088 \pm 0.048$ | 1930807 | $88.442 \pm 0.260$ |
| AGCRN | 773145 | $129.839 \pm 0.133$ | 773340 | $126.759 \pm 0.088$ | 773535 | $129.498 \pm 0.486$ |
| SCINet | 9693452 | $43.655 \pm 0.282$ | 9693476 | $45.114 \pm 0.981$ | 9693500 | $43.816 \pm 0.918$ |
| FDN | 62044 | $53.180 \pm 0.020$ | 184540 | $53.315 \pm 0.013$ | 307036 | $53.344 \pm 0.004$ |

Table 8: Total model parameters and average epoch runtime from all horizons on E-PEMS-BAY.

| Horizon | 1 | | 6 | | 12 | |
|---|---|---|---|---|---|---|
| Metric | Parameters | Runtime | Parameters | Runtime | Parameters | Runtime |
| GRU | 2753 | $3.667 \pm 0.045$ | 2753 | $4.115 \pm 0.046$ | 2753 | $4.783 \pm 0.072$ |
| TCN | 4113 | $10.938 \pm 0.144$ | 3921 | $58.752 \pm 0.942$ | 3921 | $112.923 \pm 0.320$ |
| FEDformer | 12557653 | $14.707 \pm 0.438$ | 12754261 | $15.592 \pm 0.427$ | 12950869 | $16.786 \pm 0.189$ |
| LTSF_DLinear | 8450 | $96.849 \pm 0.223$ | 50700 | $98.296 \pm 0.507$ | 101400 | $98.667 \pm 0.908$ |
| TGCN | 25985 | $23.746 \pm 0.178$ | 26310 | $23.852 \pm 0.161$ | 26700 | $23.953 \pm 0.052$ |
| A3TGCN | 62213 | $440.995 \pm 7.798$ | 62718 | $437.942 \pm 1.662$ | 63324 | $434.706 \pm 4.697$ |
| STGM | 829473 | $106.171 \pm 0.317$ | 829473 | $105.687 \pm 0.354$ | 829473 | $106.174 \pm 0.408$ |
| StemGNN | 1366452 | $39.667 \pm 0.271$ | 1366517 | $39.746 \pm 0.422$ | 1366595 | $39.521 \pm 0.081$ |
| MTGNN | 6553905 | $648.331 \pm 4.056$ | 576454 | $29.128 \pm 0.144$ | 577228 | $29.049 \pm 0.368$ |
| AGCRN | 763635 | $56.718 \pm 0.069$ | 763960 | $56.227 \pm 0.492$ | 764350 | $56.315 \pm 0.527$ |
| SCINet | 628152 | $60.520 \pm 0.758$ | 628212 | $61.030 \pm 1.108$ | 628284 | $61.053 \pm 1.274$ |
| FDN | 35150 | $36.419 \pm 0.105$ | 35470 | $36.520 \pm 0.085$ | 35854 | $36.439 \pm 0.200$ |

Table 9: Total model parameters and average epoch runtime from all horizons on Solar-Energy.

| Horizon | 1 | | 6 | | 12 | |
|---|---|---|---|---|---|---|
| Metric | Parameters | Runtime | Parameters | Runtime | Parameters | Runtime |
| GRU | 2561 | $6.687 \pm 0.299$ | 2561 | $7.021 \pm 0.065$ | 2561 | $7.751 \pm 0.098$ |
| TCN | 6097 | $10.809 \pm 0.107$ | 6097 | $49.139 \pm 0.251$ | 6097 | $94.920 \pm 0.362$ |
| FEDformer | 12381337 | $20.847 \pm 0.733$ | 12577945 | $20.850 \pm 0.057$ | 12774553 | $22.135 \pm 0.340$ |
| LTSF_DLinear | 10138 | $36.980 \pm 0.557$ | 60828 | $38.347 \pm 0.254$ | 121656 | $37.908 \pm 0.029$ |
| TGCN | 25217 | N/A | 25542 | N/A | 25932 | N/A |
| A3TGCN | 61037 | N/A | 61542 | N/A | 62148 | N/A |
| STGM | N/A | N/A | N/A | N/A | N/A | N/A |
| StemGNN | 9353900 | $31.416 \pm 0.230$ | 9354085 | $31.254 \pm 0.282$ | 9354307 | $31.409 \pm 0.159$ |
| MTGNN | 2947377 | $360.215 \pm 1.217$ | 891270 | $28.930 \pm 0.761$ | 892044 | $28.573 \pm 0.714$ |
| AGCRN | 746395 | $83.924 \pm 0.570$ | 746720 | $84.572 \pm 0.507$ | 747110 | $83.614 \pm 1.144$ |
| SCINet | 106992 | $61.431 \pm 0.153$ | 107172 | $61.577 \pm 0.234$ | 107388 | $60.910 \pm 1.350$ |
| FDN | 10654 | $35.049 \pm 0.005$ | 10974 | $35.022 \pm 0.044$ | 11358 | $34.778 \pm 0.143$ |

## A.4 MODEL EFFICIENCY RESULTS

This section covers metrics for model efficiency including total parameter counts and per-epoch runtimes. Tables 7, 8, and 9 list these metrics for Wabash River, E-PEMS-BAY, and Solar-Energy across all studied horizons. In Wabash River, we see FDN has from $1/12$ to $1/2$ as many parameters and an $\approx 2.4$ runtime speed-up compared to AGCRN. In E-PEMS-BAY, FDN has $1/38$ as many parameters as StemGNN and nearly matches it in runtime. And in Solar-Energy, FDN uses from $1/276$ to $1/78$ as many parameters as MTGNN and sees a $\approx 10.2$ speed-up in single-step forecasting but is slower for multi-step. Overall, FDN shows excellent memory and runtime performance relative to its direct competitors.

## A.5 Ablation Study Results

Tables 10, 11, 13, 12, 14, 15, and 16 present the results of all ablation studies. The best result is emboldened and the second-best is underlined. For each entry, we run three trials and present the mean result for MAE, MAPE, and RMSE. All ablation studies were conducted on the longest horizon which includes 7 days for Wabash River, 1 hour (12 time-steps) for E-PEMS-BAy, and 2 hours (12 time-steps) for Solar-Energy. Note that results for Solar-Energy in Table 11 are identical since this dataset is uni-variate.

**Learned Patterns.** The FD layer learns a set of patterns to capture $\mathbb{F}$ which, as demonstrated in Figure 1a, is greatly benefited by increasing the rank / number of patterns. Here, we test increasing the number of patterns learned by FD to improve its ability to capture $\mathbb{F}$. Table 10 and Figure 8b demonstrate the effectiveness of FD as we observe a saturation in forecast performance when learning as few as eight patterns.

**Attention Layers.** Here we test the effectiveness of LDA in FDN. We compare no attention to four attention layers of increasing specificity including (a) static attention (A) $A \in \mathbb{R}^F$ (b) dynamic attention (DA) $A \in \mathbb{R}^{I \times F}$ (c) complete localized dynamic feature attention (CLDA) $A \in \mathbb{R}^{N \times I \times F}$ and (d) our proposed LDA $A \in \mathbb{R}^{D \times I \times F}$. Results are provided in Table 11. Note that attention is not applicable to Solar-Energy since it is uni-variate. E-PEMS-BAY benefits significantly from LDA but Wabash River does not. This suggests that the importance of minimum/maximum temperature, precipitation, and soil moisture to streamflow is not specific to individual subbasins (i.e. localized).

**Dependency Embedding.** FDN encodes node inter-dependency applying Chebyshev graph convolution. Table 12 shows a significant improvement to forecast performance from the inclusion of node inter-dependency learning.

**Node Embedding Dimension.** FDN's learned node embeddings condition the classifier to each node, determine their feature filtering, and learn the graph for dependency encoding. Node embedding dimension controls the specificity of node conditioning, LDA, and the learned graph and must be tuned for proper generalization. Table 13 shows the result of increasing dimension $D$ and we observe a saturation in forecasting performance at approximately $D = 10$.

**Pattern Regularization.** In SVD, the left matrix $U$ is orthogonal to capture the highest degree of variance in $K$ column vectors. Following this, we constrain the patterns to be dissimilar by adding their similarity to forecast loss as a regularization term. Table 14 shows increasing pattern regulation, from which, Wabash River and Solar-Energy gain the most benefit.

**Periodic Moment Resolution.** Periodic embeddings consist of a sequence of $\rho$-dimensional embeddings representing moments in the known period/season of the forecast variable. Moment resolution ranges from the duration of the period ($M = 1$) to the duration of each time-step ($M \gg 1$) and must be tuned to avoid over-fitting. Table 15 shows increasing moment resolution starting from period/season duration ($M = 1$) and increasing up to time-step duration ($M{=}365$ in Wabash River). Wabash River shows saturation at 3-months (capturing the 4 seasons of the year) while E-PEMS-BAY and Solar-Energy benefit from high-resolution moments.

**Periodic Embedding Dimension.** The dimensionality of each moment embedding controls its specificity and the extent of temporal conditioning. Table 16 shows increasing periodic embedding dimension starting from $\rho{=}0$ where periodic embeddings are omitted from FDN. Periodic embeddings generally improve forecast accuracy but each dataset/signal requires a precise embedding dimension.

## A.6 High Resolution Metrics

Figures 9, 10, and 11 show node-level forecast metrics for Wabash River, E-PEMS-BAY, and Solar-Energy. These figures plot percentage change in MAPE and RMSE (where negative indicates improvement) at all nodes for FDN relative to the second-best model. The x-axis shows baseline model performance while node color intensity / size is determined by the coefficient of variation (CoV) in the forecast variable.

Table 10: Increasing the number of patterns learned by FDN.

| | Wabash River | | | E-PEMS-BAY | | | Solar-Energy | | |
|---|---|---|---|---|---|---|---|---|---|
| $K$ | MAE | MAPE | RMSE | MAE | MAPE | RMSE | MAE | MAPE | RMSE |
| 1 | 29.477 | 73.619 | 48.740 | 5.195 | 10.631 | 9.608 | 4.928 | 98.400 | 9.366 |
| 2 | 9.474 | 35.687 | 23.224 | 1.993 | 4.356 | 4.228 | 1.240 | 76.310 | 2.649 |
| 3 | 9.306 | 34.178 | 22.926 | 1.853 | 4.001 | 3.871 | 0.906 | 73.321 | 1.992 |
| 4 | 9.234 | 34.060 | 22.883 | 1.858 | 4.049 | 3.904 | 0.923 | 73.200 | 1.990 |
| 6 | 9.176 | 33.491 | 22.691 | 1.820 | 3.937 | 3.832 | 0.871 | 72.985 | 1.954 |
| 8 | 9.209 | 33.230 | 22.750 | 1.809 | 3.903 | 3.832 | 0.848 | 72.946 | **1.933** |
| 12 | 9.173 | 33.105 | 22.778 | 1.804 | 3.880 | 3.799 | 0.871 | 73.262 | 1.973 |
| 16 | 9.206 | 32.951 | 22.826 | 1.806 | 3.873 | 3.810 | 0.884 | 73.203 | 1.974 |
| 32 | **9.122** | 32.978 | **22.611** | 1.806 | 3.898 | 3.801 | **0.843** | 73.118 | 1.945 |
| 64 | 9.203 | 32.948 | 22.806 | **1.792** | 3.857 | 3.803 | 0.869 | 73.267 | 1.977 |
| 128 | 9.177 | **32.795** | 22.744 | 1.792 | **3.837** | **3.794** | 0.852 | **72.922** | 1.944 |

Table 11: Attention layers of increasing specificity including no attention (✗), static attention (A), dynamic attention (DA), full-rank localized dynamic attention (LDA), and low-rank localized dynamic attention (**LDA**).

| | Wabash River | | | E-PEMS-BAY | | | Solar-Energy | | |
|---|---|---|---|---|---|---|---|---|---|
| Attention | MAE | MAPE | RMSE | MAE | MAPE | RMSE | MAE | MAPE | RMSE |
| ✗ | **9.122** | 32.978 | **22.611** | 1.999 | 4.290 | 4.086 | **0.869** | 73.267 | **1.977** |
| A | 9.169 | 32.876 | 22.758 | 1.934 | 4.173 | 4.014 | 0.869 | 73.267 | 1.977 |
| DA | 9.190 | 33.039 | 22.836 | 1.857 | 3.983 | 3.874 | 0.869 | 73.267 | 1.977 |
| CLDA | 9.178 | **32.704** | 22.814 | 1.952 | 4.245 | 4.035 | 0.869 | 73.267 | 1.977 |
| LDA | 9.248 | 33.178 | 22.904 | **1.792** | **3.857** | **3.803** | 0.869 | 73.267 | 1.977 |

Table 12: Applying graph convolution to learn node inter-dependencies.

| | Wabash River | | | E-PEMS-BAY | | | Solar-Energy | | |
|---|---|---|---|---|---|---|---|---|---|
| GCN | MAE | MAPE | RMSE | MAE | MAPE | RMSE | MAE | MAPE | RMSE |
| ✗ | 9.321 | **32.445** | 22.982 | 1.801 | 3.828 | 3.951 | 1.042 | 74.205 | 2.370 |
| ✓ | **9.122** | 32.978 | **22.611** | **1.792** | 3.857 | **3.803** | **0.869** | 73.267 | **1.977** |

Table 13: Increasing learned node embedding dimension.

| | Wabash River | | | E-PEMS-BAY | | | Solar-Energy | | |
|---|---|---|---|---|---|---|---|---|---|
| $D$ | MAE | MAPE | RMSE | MAE | MAPE | RMSE | MAE | MAPE | RMSE |
| 1 | 9.364 | 33.075 | 23.059 | 2.077 | 4.511 | 4.266 | 0.892 | 73.452 | 2.031 |
| 2 | 9.250 | 33.269 | 22.777 | 2.097 | 4.634 | 4.329 | 0.886 | 73.165 | 2.005 |
| 3 | 9.254 | 33.178 | 22.761 | 2.014 | 4.346 | 4.173 | **0.850** | 73.170 | 1.966 |
| 4 | 9.251 | 32.927 | 22.816 | 1.967 | 4.310 | 4.106 | 0.877 | 73.361 | 1.989 |
| 6 | 9.254 | 33.099 | 22.810 | 1.861 | 4.013 | 3.895 | 0.857 | 73.261 | **1.953** |
| 8 | 9.170 | 32.809 | 22.751 | 1.818 | 3.904 | 3.838 | 0.870 | **73.013** | 1.955 |
| 10 | **9.122** | 32.978 | **22.611** | **1.792** | **3.857** | **3.803** | 0.869 | 73.267 | 1.977 |
| 12 | 9.132 | **32.766** | 22.611 | 1.821 | 3.911 | 3.840 | 0.913 | 73.265 | 2.006 |

## A.7 FORECAST SIGNAL SEASONALITY

Figures 12a, 12b, and 12c show average mutual information (AMI) of the forecast variable at each node of the system for Wabash River, E-PEMS-BAY, and Solar-Energy. The x-axis shows the time-step lag $t$ between current and past measurements for which the mutual information is calculated as $I(Y_\tau; Y_{\tau-t})$. We expect streamflow to have yearly seasonality and traffic speed and solar power to be have daily seasonality. We see this seasonality in each signal where AMI returns at a delay of 1 year ($t = 365$) for Wabash River and 1 day for E-PEMS-BAY ($t = 288$) and Solar-Energy ($t = 144$).

Table 14: Increasing regularization of learned patterns.

| | Wabash River | | | E-PEMS-BAY | | | Solar-Energy | | |
|---|---|---|---|---|---|---|---|---|---|
| $\lambda$ | MAE | MAPE | RMSE | MAE | MAPE | RMSE | MAE | MAPE | RMSE |
| 0.0000 | 9.171 | **32.623** | 22.839 | 1.792 | 3.857 | 3.803 | 0.869 | 73.267 | 1.977 |
| 0.0625 | 9.153 | 32.774 | 22.698 | **1.788** | **3.852** | **3.797** | 0.850 | 73.194 | 1.949 |
| 0.1250 | 9.172 | 32.991 | 22.634 | 1.813 | 3.929 | 3.862 | 0.849 | 73.242 | 1.957 |
| 0.2500 | 9.184 | 33.049 | 22.661 | 1.800 | 3.886 | 3.807 | 0.865 | 73.141 | 1.962 |
| 0.5000 | 9.167 | 32.847 | 22.736 | 1.819 | 3.901 | 3.824 | 0.850 | 73.253 | 1.958 |
| 1.0000 | **9.122** | 32.978 | **22.611** | 1.798 | 3.901 | 3.814 | **0.848** | **73.099** | **1.943** |

Table 15: Increasing moment resolution of learned periodic embeddings.

| Wabash River (period: yearly) | | | | E-PEMS-BAY (period: daily) | | | | Solar-Energy (period: daily) | | | |
|---|---|---|---|---|---|---|---|---|---|---|---|
| $M$ | MAE | MAPE | RMSE | $M$ | MAE | MAPE | RMSE | $M$ | MAE | MAPE | RMSE |
| 1-yr | 9.160 | 33.266 | 22.658 | 1-day | 1.786 | 3.830 | 3.792 | 1-day | 0.903 | 74.309 | 2.274 |
| 6-mon | 9.213 | 33.194 | 22.613 | 12-hr | 1.787 | 3.795 | 3.776 | 12-hr | 0.940 | 73.498 | 2.091 |
| 3-mon | **9.122** | 32.978 | **22.611** | 6-hr | 1.792 | 3.857 | 3.803 | 6-hr | 0.869 | 73.267 | 1.977 |
| 1-mon | 9.140 | **32.941** | 22.702 | 3-hr | 1.806 | 3.886 | 3.796 | 3-hr | 0.857 | 73.076 | 1.950 |
| 7-day | 9.166 | 33.025 | 22.815 | 1-hr | 1.775 | 3.818 | 3.756 | 1-hr | **0.841** | 72.898 | **1.922** |
| 1-day | 9.336 | 33.197 | 23.063 | 5-min | **1.757** | **3.794** | **3.733** | 10-min | 0.851 | **72.889** | 1.945 |

Table 16: Increasing learned periodic embedding dimension. At dimension 0, periodic embeddings are omitted from FDN entirely.

| | Wabash River | | | E-PEMS-BAY | | | Solar-Energy | | |
|---|---|---|---|---|---|---|---|---|---|
| $\rho$ | MAE | MAPE | RMSE | MAE | MAPE | RMSE | MAE | MAPE | RMSE |
| 0 | 9.181 | 32.802 | 22.718 | 1.866 | 4.047 | 3.920 | 0.899 | 74.308 | 2.257 |
| 1 | **9.122** | 32.978 | **22.611** | 1.792 | 3.857 | 3.803 | 0.869 | 73.267 | 1.977 |
| 2 | 9.229 | 33.087 | 22.807 | 1.836 | 3.977 | 3.877 | 0.876 | 73.201 | 1.977 |
| 4 | 9.173 | 33.236 | 22.676 | **1.773** | **3.803** | **3.758** | 0.895 | 73.188 | 2.000 |
| 8 | 9.204 | **32.736** | 22.870 | 1.802 | 3.896 | 3.775 | **0.864** | **72.989** | **1.955** |

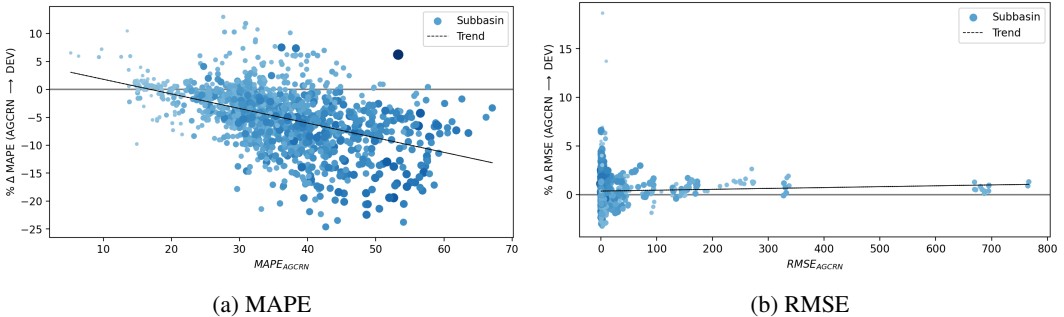

(a) MAPE                (b) RMSE

Figure 9: Distribution of forecast error between AGCRN and FDN across subbasins of the Wabash River. Values are given as a percentage change where negative indicates a reduction in forecast error by FDN.

## A.8 FDN PARAMETER SETTINGS

Here we cover all settings of FDN on each dataset for reproducibility. Table 17 provides the model parameter settings for each dataset. Parameters shared across all datasets include (a) a 200 epoch limit (b) a patience of 15 epochs (c) mini-batch size 64 (d) the Adam optimizer (Kingma & Ba, 2014) (e) a learning rate of 0.003 (f) Kaiming normal initialization (He et al., 2015) and (g) L1Loss as forecast loss.

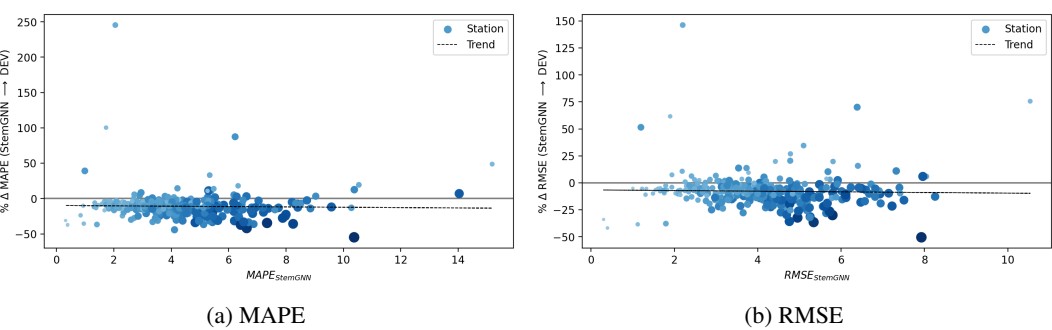

(a) MAPE  (b) RMSE

Figure 10: Distribution of forecast error between StemmGNN and FDN across stations of E-PEMS-BAY. Values are given as a percentage change where negative indicates a reduction in forecast error by FDN.

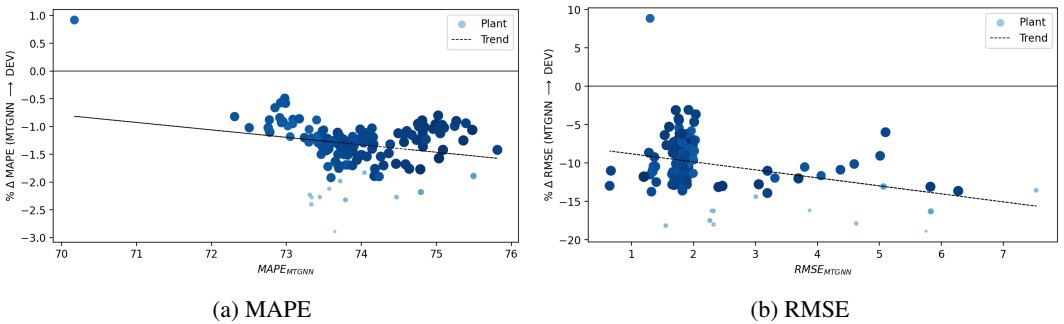

(a) MAPE  (b) RMSE

Figure 11: Distribution of forecast metrics between MTGNN and FDN across plants of Solar-Energy. Values are given as a percentage change where negative indicates a reduction in forecast error by FDN.

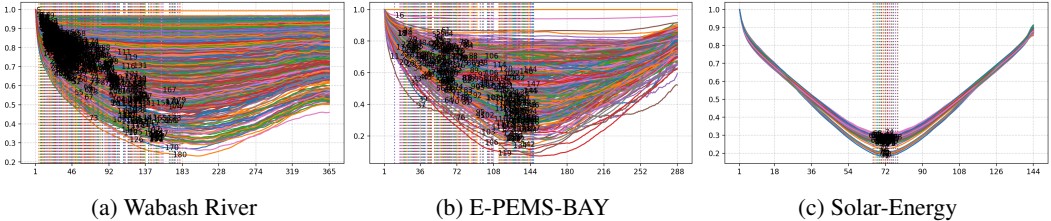

(a) Wabash River  (b) E-PEMS-BAY  (c) Solar-Energy

Figure 12: Average mutual information of the forecast variable for each node in the dataset. The x-axis indicates time-step lag between $x$ and $y$ when calculating $I(x; y)$.

Table 17: Model parameter settings for each dataset.

| Dataset | $K$ | $H$ | Attn | $D$ | $\lambda$ | $M$ | $\rho$ |
|---|---|---|---|---|---|---|---|
| Wabash River | 32 | 32 | ✗ | 10 | 1.0 | 3-months | 1 |
| E-PEMS-BAY | 64 | 64 | **LDA** | 10 | 0.0 | 6-hours | 1 |
| Solar-Energy | 64 | 64 | N/A | 10 | 0.0 | 6-hours | 1 |

