# OpenReview forum: "FDN: Interpretable Spatiotemporal Forecasting with Future Decomposition Networks"
_ICLR.cc/2025/Conference — ICLR 2025 Conference Withdrawn Submission_

### Official Review · Reviewer_Z8HQ · 2024-11-01

**Soundness:** 3
**Presentation:** 2
**Contribution:** 3
**Rating:** 5
**Confidence:** 2

**Summary:**

The article presents an intriguing spatiotemporal prediction model that defines a learnable pattern matrix, which calculates the weights of various patterns based on the input to yield interpretable predictions. Experimental results indicate that this predefined matrix successfully captures temporal pattern information and achieves strong predictive performance across various benchmarks.

**Strengths:**

- The article introduces an intriguing prediction mechanism that allows the model to autonomously learn temporal patterns and combine them to generate predictions, thereby enhancing interpretability.
- Experimental results demonstrate that the model exhibits strong predictive performance across three datasets. Additionally, by visualizing the pattern matrix $F$ , it is evident that the model has indeed learned certain temporal patterns, contributing to the interpretability of its predictions.

**Weaknesses:**

- **Insufficient Novelty of the Method:** Specifically, the primary contribution of the paper is the transformation of the spatiotemporal prediction task into a matrix factorization task, which has already been explored in previous literature [1].
- **Unclear Method Description:** The rationale for why the two learnable matrices $E$ $E$$P$ can respectively represent spatial and periodic information is not clearly articulated in the paper; aside from gradient descent, there are no explicit constraints provided.
- **Lack of Clarity in Writing:** The overall writing of the article requires improvement, as reviewers may struggle to intuitively grasp the main content and contributions of the paper.

**Reference**
[1] Yu, H. F., Rao, N., & Dhillon, I. S. (2016). Temporal regularized matrix factorization for high-dimensional time series prediction. Advances in neural information processing systems, 29.

**Questions:**

- The authors have used a limited number of test datasets. Could you add more datasets (like PEMS-04, PEMS-03, and Traffic) to make the experimental results more convincing?
- The authors should clarify how the two learnable matrices EEE and PPP can learn spatial and periodic information.
- From Figure 8, it appears that as the number of predefined patterns increases, the model's performance improves until it reaches saturation. However, I believe that having too many predefined patterns may cause the model to overfit local patterns, leading to a decrease in performance. Can the authors explain this?

---

### Official Review · Reviewer_dU6X · 2024-11-03

**Soundness:** 4
**Presentation:** 2
**Contribution:** 2
**Rating:** 6
**Confidence:** 4

**Summary:**

This paper presents a new method: Future Decomposition Network (FDN) for multivariate time series forecasting. FDN frames the problem of forecasting into two stages:
1. Classifying the input $X$ into $K$ types of future patterns seen in training data (this is done for each node).
2. Interpolating the final prediction $\boldsymbol{Y}$ based on the likelihood vector (also done for each node).

To get the likelihood vector, an "encoder" is used to produce node embeddings with additional structural and temporal information. Structural information is added with learnable node embeddings and processed with GCN for structure-aware message passing. Temporal information is added with periodic embeddings and processed with GRU.
The prediction layer, Future Decomposition Layer (FDL), is a learnable embedding matrix that aims to summarize all possible dynamics for a node. This layer provides interpretability as we can directly visualize the favourable predictions.

The authors conduct experiments on 3 different spatiotemporal forecasting datasets and show improvements over existing methods in 3 metrics for 3 prediction horizons. They also demonstrate the interpretability of the model by showing the likely patterns of FDL used in prediction and visualizing the low-dimensional space of the learned matrix.

Overall, the technical contribution of the paper is in framing forecasting as a "classification" problem that increases interpretability and efficiency.

**Strengths:**

To the best of my knowledge, the paper is the first to formulate the problem of spatiotemporal forecasting as a classification and interpolation process. In doing so, the authors created a new framework that is very interpretable, memory-efficient, and accurate.

At the sentence level, the writing is mostly clear and easy to follow. The figures not only help readers understand the architecture and information flow (Figures 2, 3, 4) but also give readers an intuitive understanding of what the model is learning and how it is producing the predictions (Figures 1, 7, 8).

The empirical prediction improvements are also impressive and siginifcant.

**Weaknesses:**

* **Literature review and baseline comparisons**:

    * The paper lacks literature reviews on neural memory networks. For instance, PM-MemNet by Lee et al. [1] matches input traffic data to representative patterns with a key-value memory structure for forecasting. While the model architecture is different, the authors should clarify the differences between PM-MemNet and FDN.

    * The authors should also note that the technical details in Localized Dynamic Attention module (using node embeddings $\boldsymbol{E}$ and weights $W$ to produce $\hat{A}$) are very similar to the Node Adaptive Parameter Learning module in AGCRN by Bai et al. [2]. While the overall architecture is different, the authors should address this module-level inspiration since the paper is already mentioned in related works.

    * For baseline comparisons, the authors should also include DCRNN [3] and STGCN [4] due to their significance. However, this does not affect the score. MMR-GNN [5] is also mentioned as a dataset source, but it is not included as a baseline comparison.

* **Experimental details**:
    * The effectiveness of concatenating the node embeddings to the input is unclear and should be examined in ablation studies, similar to how the model was evaluated without periodic embeddings.
    * The authors should include anonymized code for reproducibility (does not affect score)
    * It would be interesting to see if the most important patterns learned by $\hat{\mathbb{F}}$ are similar to the patterns of $\mathbb{F}$ presented in Figure 1.b.

* **Writing clarity**:
    * In section 3.2.2, the paper introduces many notations that are not defined prior to use. For instance, when explaining the dimensions of the matrices, $D, \rho$ are explained later, and $F, H$ are never explained (I am assuming they are hidden dimensions). It could be better to move equation 3 in preamble classification to the last section, after explaining the individual components.
    * $O$ is being overloaded as both the dimension in $\mathbb{F}$ and the prediction horizon $\boldsymbol{Y}$.
    * Figure 12 is very hard to read


* **Typos**
    * Line 461-462 4:30pm >> 4:30 p.m.
    * Line 465-466 Here we observe >> Here, we observe
    * Line 299 $A$ is not defined, maybe typo
    * Many references are from ArXiv, not from their original published journals (e.g. Lines 559-563). Also, the "Attention is all you need paper" (Line 604-605) is missing other authors.

1. Lee, H., Jin, S., Chu, H., Lim, H., & Ko, S. (2021). Learning to remember patterns: pattern matching memory networks for traffic forecasting. arXiv preprint arXiv:2110.10380.
2. Bai, L., Yao, L., Li, C., Wang, X., & Wang, C. (2020). Adaptive graph convolutional recurrent network for traffic forecasting. Advances in neural information processing systems, 33, 17804-17815.
3. Li, Y., Yu, R., Shahabi, C., & Liu, Y. (2017). Diffusion convolutional recurrent neural network: Data-driven traffic forecasting. arXiv preprint arXiv:1707.01926.
4. Yu, B., Yin, H., & Zhu, Z. (2017). Spatio-temporal graph convolutional networks: A deep learning framework for traffic forecasting. arXiv preprint arXiv:1709.04875.
5. Majeske, N., & Azad, A. (2024, April). Multi-modal Recurrent Graph Neural Networks for Spatiotemporal Forecasting. In Pacific-Asia Conference on Knowledge Discovery and Data Mining (pp. 144-157). Singapore: Springer Nature Singapore.

**Questions:**

* Overall, how is FDN different from the goal of $K$-means clustering or Gaussian Mixture Models? Can we use these unsupervised learning approaches to learn $\mathbb{F}$, instead of learning it through classification? Would it not be better to learn the distribution this way?
* Mathematically, the Future Decomposition Layer is a set of learnable weights used during matrix multiplication for prediction. How is this different from a fully connected layer without bias? If so, should it be categorized as such, similar to other models that use linear prediction layers? On the other hand, if other models "force" the size of the prediction layer to be $K\times O$, would the prediction layer be more interpretable?
* GCN implementation details were unclear. Specifically, how many layers are used, how is $\boldsymbol{E}$ used in the graph convolution process?

---

### Official Review · Reviewer_SFbb · 2024-11-04

**Soundness:** 1
**Presentation:** 2
**Contribution:** 1
**Rating:** 3
**Confidence:** 5

**Summary:**

The article proposes the Future Decomposition Network (FDN) for spatiotemporal prediction, utilizing classified information to provide more interpretable forecasts based on latent spatiotemporal patterns. The effectiveness of FDN is demonstrated across hydrologic, traffic, and energy system datasets, showcasing its potential for enhancing predictive accuracy and interpretability in various domains.

**Strengths:**

1. FDN provides a certain level of interpretability for spatiotemporal predictions.
2. Experiments were conducted on three types of datasets, including datasets with additional features, along with ablation and hyperparameter experiments. Furthermore, multiple experiments reported the expected values and variances.

**Weaknesses:**

To assist in enhancing the quality of the research paper, I believe there are areas that could be improved as outlined below:
1. The novelty claimed in this paper is not convincing to me, as the operation of weighting and summing a set of fundamental patterns shared by all entities, as mentioned in line 45 has been extensively studied [1]. Even the configuration of modules in the Classifier is just a combination of common modules in current spatiotemporal models, without demonstrating the benefits of this particular setup.
2. The authors did not clearly explain the significance of interpretability and why existing spatiotemporal models lack interpretability. Why can weighting the sum of a given number of patterns be considered interpretable? The authors also state in line 159, "Only attention provides direct interpretability through the examination of final attention scores." Therefore, at least models utilizing attention in traditional spatiotemporal modeling are claimed to have interpretability. I am puzzled by the fact that the authors claim interpretability comes from the weighted sum of given patterns, yet the process of obtaining these weights is through a series of deep models, meaning that the learning of these weights does not possess interpretability.
3. In the abstract and in line 18, the authors mention that FDN features "delivering forecasts competitive with SOTA methods at a fraction of their memory and runtime cost." However, the authors did not provide an explanation or illustration of the model's training/inference speed and memory usage compared to baselines. Merely comparing parameters is not a strong evidence, as in practice, memory usage (or foot-print) is the primary factor affecting model deployment.
4. I am concerned about the performance of the model, as it lacks against SOTA baselines. Additionally, the datasets used are not commonly recognized in spatiotemporal learning. Furthermore it is necessary to test on common spatiotemporal datasets, such as PeMS0X (X=3, 4, 7, 8) and it is recommended to test on larger spatiotemporal datasets, including LargeST [2] and XTraffic [3], with thousands to tens of thousands of nodes to demonstrate the advantages of the model in terms of memory and runtime cost and to validate the model's efficiency.
5. The authors validated the model's effectiveness in multiple scenarios, including hydrologic, traffic, and energy systems, all of which are stated to have stable periodic information in the original text. I recommend adding scenarios where the periodic information is less obvious, such as the KnowAir dataset [4] and the datasets in GAGNN [5] in air quality prediction scenarios , to further ensure the model's effectiveness in more complex spatiotemporal feature backgrounds.
6. The experiments lack comprehensive baselines: the authors did not compare with classical models similar to their methods, such as PM-MemNet [1], and lack some classical but powerful spatiotemporal models, such as Graph WaveNet [6], STID [7], D$^2$STGNN [8], as well as the latest unavoidable SOTA models emerging in spatiotemporal research, such as STONE [9] and BigST [10]. These models focus on interpretability, performance, or efficiency, and it is inevitable to compare them with the authors' emphasis on FDN having interpretability, high efficiency, and high performance.
7. Regarding hyperparameter experiments, I observed that some metrics in the author-provided bound parameter settings are still optimal. For example, in Table 10, when $K=128$, the model's MAPE in Wabash River and Solar-Energy, as well as the MAPE and RMSE in E-PEMS-BAY, are still optimal. The authors need to further increase the number of $K$ to demonstrate that the current hyperparameter selection is the optimal choice, meaning that further increasing this hyperparameter does not lead to further improvements in the model. This issue of lack of further validation in hyperparameter experiments is evident in Table 10, 13-16.
8. Lastly, the authors generate the final result by weighting and summing the given patterns through softmax probability based on the last FC result in the Classifier. I am more curious about whether disregarding the predefined patterns and treating the Classifier as a 'Regressor', with the final FC output dimension changed to $N\times T$ to directly serve as the result, would yield better performance, setting aside the interpretability background claimed by the authors. This could be considered as a new ablation experiment demand.

[1] Lee H, Jin S, Chu H, et al. Learning to remember patterns: pattern matching memory networks for traffic forecasting[J]. arXiv preprint arXiv:2110.10380, 2021.
[2] Liu X, Xia Y, Liang Y, et al. Largest: A benchmark dataset for large-scale traffic forecasting[J]. Advances in Neural Information Processing Systems, 2024, 36.
[3] Gou X, Li Z, Lan T, et al. XTraffic: A Dataset Where Traffic Meets Incidents with Explainability and More[J]. arXiv preprint arXiv:2407.11477, 2024.
[4] Wang S, Li Y, Zhang J, et al. Pm2. 5-gnn: A domain knowledge enhanced graph neural network for pm2. 5 forecasting[C]//Proceedings of the 28th international conference on advances in geographic information systems. 2020: 163-166.
[5] Chen L, Xu J, Wu B, et al. Group-aware graph neural network for nationwide city air quality forecasting[J]. ACM Transactions on Knowledge Discovery from Data, 2023, 18(3): 1-20.
[6] Wu Z, Pan S, Long G, et al. Graph wavenet for deep spatial-temporal graph modeling[J]. arXiv preprint arXiv:1906.00121, 2019.
[7] Shao Z, Zhang Z, Wang F, et al. Spatial-temporal identity: A simple yet effective baseline for multivariate time series forecasting[C]//Proceedings of the 31st ACM International Conference on Information & Knowledge Management. 2022: 4454-4458.
[8] Shao Z, Zhang Z, Wei W, et al. Decoupled dynamic spatial-temporal graph neural network for traffic forecasting[J]. arXiv preprint arXiv:2206.09112, 2022.
[9] Wang B, Ma J, Wang P, et al. Stone: A spatio-temporal ood learning framework kills both spatial and temporal shifts[C]//Proceedings of the 30th ACM SIGKDD Conference on Knowledge Discovery and Data Mining. 2024: 2948-2959.
[10] Han J, Zhang W, Liu H, et al. BigST: Linear Complexity Spatio-Temporal Graph Neural Network for Traffic Forecasting on Large-Scale Road Networks[J]. Proceedings of the VLDB Endowment, 2024, 17(5): 1081-1090.

**Questions:**

I believe the authors are trying to emphasize that the performance and interpretability come from combining fixed patterns according to specific weights. Therefore, the Classifier is not a focal point. I am also curious whether, for a given set of $K$ patterns, all spatiotemporal models, by changing the final output dimension from $N\times T$ to $N\times K$ and generating weights for corresponding patterns through softmax probability, would achieve similar or better results. In essence, is this operation a universally enhancing operation?

Others see weakness.

---

### Official Review · Reviewer_mUa8 · 2024-11-05

**Soundness:** 3
**Presentation:** 2
**Contribution:** 2
**Rating:** 5
**Confidence:** 4

**Summary:**

This paper addresses the problem of spatio-temporal forecasting using a decomposition-based neural network. The major idea is to reveal latent patterns of the predictive targets. The experiments are conducted over three real-world datasets.

**Strengths:**

1. Spatio-temporal forecasting is not only practical but also carries significant social implications.
2. The insights presented in this paper are both compelling and well-founded.
3. The proposed model demonstrates efficiency (as evidenced in A.4) and offers a degree of interpretability.

**Weaknesses:**

1. Although the insight is interesting,  the technical contributions to the ICLR community appear to be relatively limited in scope.

2. The experiments could be enhanced in the following ways:
a) The choice of baselines is somewhat outdated, which weakens the comparability of the model's performance (particularly in Table 2). To the best of my knowledge, a significant body of research on spatio-temporal forecasting and STGNNs has been published in prestigious venues [1] such as KDD, NeurIPS, ICML, ICLR, and AAAI/IJCAI. It would be valuable to consider incorporating some of these works for a more robust comparison.
b) It would be beneficial to evaluate the model on more widely used spatio-temporal forecasting datasets, such as PEMS-03/04/07/08, LargeST, and METR-LA. I am particularly interested in understanding the model's performance improvements on these mainstream datasets.

3. The writing and literature review could be further improved. For instance, the pipelines depicted in Figures 3 and 4 are not sufficiently explained and could benefit from clearer illustrations. Additionally, a more comprehensive literature review is needed, particularly focusing on pattern-matching neural networks in time series or spatio-temporal forecasting (e.g., [2, 3]). It would be helpful to highlight the major differences between the proposed model and existing approaches, as well as to clarify the key advantages of the proposed method in comparison.

Reference:

[1] Spatio-Temporal Graph Neural Networks for Predictive Learning in Urban Computing: A Survey. TKDE 2023.

[2] Learning to Remember Patterns: Pattern Matching Memory Networks for Traffic Forecasting. ICLR 2022.

[3] Multi-scale Traffic Pattern Bank for Cross-city Few-shot Traffic Forecasting. arXiv.

**Questions:**

1. Does the paper have results on zero-shot or few-shot forecasting? If the training dataset is limited, could we learn useful patterns that can be generalized to the unseen data?

2.  What's the major difference/advantage of the proposed model against existing pattern-matching methods for TS/ST forecasting?

---

### Note · Authors · 2024-11-24

**Comment:**

The lead author of the paper has a family emergency. Hence, we are unable to submit a formal rebuttal to address reviewers' questions.

**Withdrawal Confirmation:**

I have read and agree with the venue's withdrawal policy on behalf of myself and my co-authors.